# Scalable Diverse Model Selection for Accessible Transfer Learning

**Daniel Bolya**[*]
Georgia Tech
dbolya@gatech.edu

**Rohit Mittapalli**[*]
Georgia Tech
rmittapalli3@gatech.edu

**Judy Hoffman**
Georgia Tech
judy@gatech.edu

## Abstract

With the preponderance of pretrained deep learning models available off-the-shelf from model banks today, finding the best weights to fine-tune to your use-case can be a daunting task. Several methods have recently been proposed to find good models for transfer learning, but they either don't scale well to large model banks or don't perform well on the diversity of off-the-shelf models. Ideally the question we want to answer is, "given some data and a source model, can you quickly predict the model's accuracy after fine-tuning?" In this paper, we formalize this setting as "Scalable Diverse Model Selection" and propose several benchmarks for evaluating on this task. We find that existing model selection and transferability estimation methods perform poorly here and analyze why this is the case. We then introduce simple techniques to improve the performance and speed of these algorithms. Finally, we iterate on existing methods to create PARC, which outperforms all other methods on diverse model selection. We have released the benchmarks and method code[†] in hope to inspire future work in model selection for accessible transfer learning.

## 1 Introduction

Deep Neural Networks (DNNs) have shown to be very capable of solving a wide variety of visual tasks. However, these networks often require large amounts of data and training time to perform well, limiting the *accessibility* of deep learning for computer vision. One approach to alleviate this problem is to employ transfer learning, commonly by fine-tuning an off-the-shelf model on the desired task.

With the increasing number of off-the-shelf models available spanning different tasks, datasets, training methods, and architectures, choosing the best model from which to transfer is a challenging endeavor. A common heuristic in computer vision has been to use models pretrained on ImageNet [7] (specifically the ILSVRC challenge data), but more recent work is starting to expose weaknesses in the generalization performance of ImageNet features [39, 28, 40], and many of these works find that other pretraining datasets may perform better for some target tasks. Aside from source dataset, which architecture to select isn't clear (when comparing similarly fast models). One could theoretically train several transfers from a diverse set of pretrained weights and keep the best performing model, but this isn't feasible in most practical applications where compute is limited.

This motivates the need for a model selection method that could query a large bank of existing pretrained models (of which several already exist, e.g., [36, 56, 18]) with a small subset of the target data and return a set of weights which performs well when fine-tuned on all target data. Because of the sheer quantity of pretrained weights available to download off the internet today, such a model bank has the potential to be massive and cover a wide variety of tasks, datasets, and architectures. Any

---

[*]Equal Contribution

[†]https://dbolya.github.io/parc/

35th Conference on Neural Information Processing Systems (NeurIPS 2021).

selection method that intends to operate on such a massive library of weights would thus need two important properties: it would need to be *scalable* in order to accommodate 100s to 1000s of source models, and it would need to perform well on a *diverse* set of input weights due to the wide variety of models available. In this paper, we formalize this setting as *Scalable Diverse Model Selection*.

Currently, there are two main lines of work that could address this scenario. The first is model selection [11, 46, 10], which attempts to select a viable transfer from a suite of arbitrary pretrained models. However, existing approaches require an initial model trained on the target data to suggest the transfer (which limits accessibility) and are not very successful when comparing across architectures (see Tab. 1). The second line of work is in transferability estimation [4, 52, 34], which attempts to predict how well a source model will transfer to a target task. While this might seem similar to model selection, the two tasks are subtly different in evaluation. Transferability estimation methods usually fix one source model and vary the target task (i.e., "For my source model, which task would it transfer to the best?"), while model selection methods do the opposite: fix a target task and vary the source models (i.e., "For my target task, which source model will transfer the best?"). While it might seem the same methods could work for either case, this turns out to not be the case (see Tab. 2).

**Contributions.** We formalize the task of *Scalable Diverse Model Selection*, which intends to make deep learning for computer vision more accessible. While other papers might have explored aspects of this space already, we standardize it by introducing several tools and benchmarks for evaluating model selection methods in this setting. First, we provide a controlled environment that includes exhaustively trained transfers from 8 source datasets to 6 target datasets across 4 different commonly used architectures for a total of 168 ground truth transfers for analysis (Sec. 3). We show that current state-of-the-art transferability and model selection methods fail to beat simple baselines in this new setting (Tab. 1). We then analyze why this is the case and provide techniques to improve performance (Sec. 4). Using insights from this analysis, we develop PARC, a method that outperforms other methods on this benchmark (Sec. 5). Finally, we show that these results generalize to a larger experiment with an extra dataset and 33 additional off-the-shelf pretrained models downloaded from the internet (for a total of 65 source models and 423 transfers) and by extending PARC to object detection (Sec. 6). We have released all benchmarks and evaluation code at `https://dbolya.github.io/parc/` in hopes to further the development of this promising area of research.

## 2 Related Work

In this paper, we introduce a new task for Scalable Diverse Model Selection, which attempts to make transfer learning more accessible by selecting the best pretrained model for a downstream task from a massive bank of off-the-shelf models. This is adjacent to several fields such as accessible transfer learning, transferability estimation, and Taskonomy model selection.

**Transfer Learning.** Transfer learning is the act of using a model trained in one setting to boost the performance or speed up the training of a model in another setting, and is an extensively studied field in deep learning [49]. Several approaches exist for transfer learning from a source model to a target dataset, the most popular of which include fine-tuning a new head to the target domain [9, 43, 60, 39], and fine-tuning the entire network [2, 17]. While there are more sophisticated methods for fine-tuning (e.g., [19, 37]), we study fine-tuning the entire network, as it's simple and widely adopted.

**Accessible Transfer Leaning.** A few works have tried to make transfer learning more accessible. Neural Data Server [57] allows users to augment their own data with similar data indexed from several massive image datasets. While helpful in a limited data environment, this requires the user to have additional compute in order to train with the extra images. Iterating on this idea, Scalable Transfer Learning with Expert Models [38] suggests pretrained models to the user instead, relaxing the compute requirement. However, the focus of the paper is primarily on creating these pretrained "expert" models and not in selecting between them. We argue that there are several "expert" models already widely available and trained with a large amount of data (e.g., [47, 33, 39]), and thus focus on the model selection aspect of this problem. Both of these works use simple baselines to guide their model selection: Neural Data Server [57] uses the accuracy of a logistic classifier on the source features fit to the target task, while Scalable Transfer Learning [38] uses nearest neighbors with hold-one-out cross-validation. We benchmark both baseline techniques in our setting.

**Transferability Estimation.** There are several works that attempt to predict how well a model will transfer to new tasks, ranging from methods that attempt to assess a models capacity for transfer [23, 28, 40] to those that attempt to predict transfer accuracy [12, 3, 45]. Others attempt to predict the gap in generalization between training and test time [24, 53, 25]. There has also been a recent line of work that directly attempts to estimate transfer learning accuracy given a source model and target dataset [4, 52, 34]. This line of work is the most applicable to our setting because the only assumption they make on the source model is that it was trained on classification. While not ideal, this allows us to benchmark these methods (H-Score [4], NCE [52], and LEEP [34]) in our setting. Otherwise, Deshpande et al. [8] propose perhaps what is most directly applicable to our work. Their setting is very similar (and fairly concurrent), though not as diverse and with no restrictions on evaluation speed. In addition, LogMe [58] is concurrent work that also focuses on practical transferability estimation, but they don't release their benchmark and their setting is more narrow.

**Taskonomy Model Selection.** The Taskonomy [59] models and dataset has been used to benchmark a previous line of model selection algorithms [11, 46, 10]. Taskonomy attempts to model similarities between tasks by how well models transferring from one task to another perform after fine-tuning. This makes it a natural test-bed for model selection evaluation, as it contains a large number of pretrained transfers to use as a benchmark. However, using Taskonomy as a benchmark is incomplete. All Taskonomy models are trained on the same data (with different labels) and follow roughly the same architecture, only varying the source and target task. This means that the benchmark doesn't test robustness to source model *diversity*, which is a core tenant of this work. Furthermore, typical model selection works on Taskonomy require a network trained on the target data to suggest transfers, which should be avoided to make transfer learning more accessible. We benchmark the performance of RSA [11] and DDS [10], as they are the best performing methods on Taskonomy model selection. There are other works in this space such as [6] and [1], but the former scales poorly with the number of sources, and the latter cannot compare across architectures by design, so we don't include either in our experiments.

# 3 Scalable Diverse Model Selection

In this work, we address the problem of model selection for transfer learning but through an expressly practical lens. The goal of model selection from a practitioner's point of view is to find a pretrained off-the-shelf model that will perform well after fine-tuning on their data. In order for a model selection method to perform well in this setting, the models it selects from need to be *diverse* (i.e., cover a wide variety of source datasets, architectures, and pretraining tasks), and the selection method needs to be *scalable* (since the number of off-the-shelf models available today is massive and growing).

In order to provide a realistic transfer learning model selection scenario, we propose to select a model from a *large* source model bank (over 100 models spanning several datasets and architectures) that transfers well to a target training set after full model fine-tuning. The naive approach to this problem would be to simply fully fine-tune a transfer from each source model to the target training set, but this would of course be computationally infeasible. We'd like to work in the practical setting where we don't train any extra models, so we'd like a *computationally efficient* transferability estimation method to predict which source models would transfer well. For the same reason, it's also infeasible to use the entire target training set for this estimate, since extracting all the features for 100 different source models is akin to training a new model for 100 epochs (though without the backward pass). Thus, we restrict the method to only use a small subset of the target training data for its estimate.

In this scenario, the target dataset is fixed while the source models can vary in dataset, architecture, and task. Previous work in model selection and transferability estimation typically only vary one of these aspects when evaluating their method (i.e., just task [59, 4, 11, 10], just dataset [34, 52], or just architecture [34], but not all three at once). It's unclear how well, if at all, these methods benchmarked by varying only one factor will perform when all of these factors can change. Thus, we've set out to test these methods in this much more challenging setting.

## 3.1 Creating a Diverse Benchmark

Because no model selection benchmark currently exists that varies more than one source factor at time, we create our own from 8 source datasets and 6 target datasets across 4 different architectures for classification, varying both source *dataset* and *architecture*. We will also vary task in Sec. 6.

**Datasets and Architectures.** An ideal selection of source datasets would contain related datasets, so that transfer learning makes sense. Thus, for this benchmark, we choose 6 well-known classification datasets of various difficulties that contain related subthemes: **Pets**: Stanford Dogs [26] and Oxford Pets [35], **Birds**: CUB200 [55] and NA Birds [54], and **Miscellaneous**: CIFAR10 [29] and Caltech101 [14]. We also include VOC2007 [13] and ImageNet 1k [7] as the 7th and 8th source datasets, but not as targets. The other 6 datasets are also included as targets.

When selecting a model, a practitioner is often interested in the accuracy-speed trade-off, meaning that the benchmark should include architectures at several tiers of evaluation speed. To facilitate this, we include three tiers of architectures: ResNet-50 [22] as the slowest, ResNet-18 [22] and GoogLeNet [48] in the middle, and AlexNet [30] as the fastest.

**Evaluation.** The goal in model selection is to find a source model that will transfer well to a target task. Ideally, this would be the best performing model, but that might be unreasonably difficult when there are 100s of models to choose from. Furthermore, practitioners are typically interested in the trade-off between performance and inference speed, which requires us to consider more than just the highest scoring models. What we really need in this case is a score for each model that correlates well with final fine-tuned accuracy on some target data. To benchmark such a score, we use Pearson Correlation [15], a widely adopted correlation metric (with 0 implying no correlation and 100 implying perfect correlation), between the model selection algorithm's transferability scores and the final fine-tuned accuracy.

Thus we employ the following procedure to test a model selection method $\mathcal{A}$ on our benchmark: for each target dataset $\mathcal{D}^t$ (indexed by $t \in T$), we sample an $n$ image "probe set" $\mathcal{P}_n^t \subseteq \mathcal{D}^t$. Then, for each source model parameterized by $\theta_s$ (indexed by $s \in S$), we obtain

$$\alpha_s^t = \mathcal{A}(\theta_s, \mathcal{P}_n^t) \tag{1}$$

as the predicted score for how well the source model $\theta_s$ will transfer to the target dataset $\mathcal{D}^t$. Then, we train each transfer from $\theta_s$ to $\mathcal{D}^t$ and evaluate it on the test set of $t$ to obtain the final transfer accuracies $\omega_s^t$. Finally, we compute Pearson correlation (denoted by `pearsonr`) between the predicted and final transfer accuracy and average it over all target datasets:

$$\text{Mean PC (Varying Source)} = \frac{1}{|T|} \sum_{t \in T} \text{pearsonr}(\{\alpha_s^t : s \in S\}, \{\omega_s^t : s \in S\}) \tag{2}$$

Because there can be a large amount of variance in the probe set sampled, we further report mean and standard deviation over 5 different randomly sampled probe sets.

Note that we explicitly use Pearson Correlation because it incorporates all data points, if all you care about is selecting the most accurate model (without limits on speed or other model parameters), other metrics such as top-k accuracy may be more suitable. Thus, we include some extra metrics in the Appendix.

**Implementation Details.** For simplicity, images from all datasets are resized to $224 \times 224$. When constructing the probe sets, we ensure that there are at least 2 examples of each class (necessary for several methods) and randomly subsample classes if this results in more than $n = 500$ images. We train all source models and transfers using SGD with no weights frozen (i.e., full fine-tuning) and employ grid search to find optimal hyperparameters for each target dataset, architecture pair. Previous work in this space assume that each target model is trained with exactly the same hyperparameters. However, in a practical setting we expect some tuning to be done when training on the target dataset, so we have done the same. This makes the selection task more challenging but also more aligned with best case transfer outcomes. All models are trained on Titan Xp GPUs and all transferability methods are evaluated on the CPU. Note that times should be taken as lower bounds, since they're evaluated with expensive hardware. Many would-be practitioners don't have such hardware at their disposal.

## 3.2 Benchmarking Existing Work

We use this benchmark to evaluate several recent model selection and transferability estimation methods. We also include some typical baselines to contextualize the performance of these methods.

**Probability-Based Methods.** We test two very recent transferability estimation methods, NCE [52] and LEEP [34]. Both operate similarly: given a source model parameterized by $\theta$ that predicts

| Method | Input | Training Time | Source Task Agnostic | Target Task Agnostic | Mean PC (% ↑) | Time (ms ↓) |
|---|---|---|---|---|---|---|
| NCE [52] | $p_\theta(z \mid x), y$ | | | | $2.1 \pm 0.7$ | $3.3 \pm 4.6$ |
| LEEP [34] | $p_\theta(z \mid x), y$ | | | | $10.8 \pm 0.1$ | $3.4 \pm 4.3$ |
| H-Score [4] | $f_\theta(x), y$ | | ✓ | | $-5.4 \pm 4.9$ | $5049.3 \pm 7200.2$ |
| RSA AlexNet [11] | $f_\theta(x)$ | 1 Hour | ✓ | ✓ | $-1.4 \pm 0.6$ | $93.7 \pm 19.1$ |
| DDS AlexNet [10] | $f_\theta(x)$ | 1 Hour | ✓ | ✓ | $1.7 \pm 0.4$ | $65.8 \pm 17.9$ |
| RSA ResNet-50 [11] | $f_\theta(x)$ | 3 Hours | ✓ | ✓ | $57.3 \pm 0.4$ | $102.7 \pm 15.6$ |
| DDS ResNet-50 [10] | $f_\theta(x)$ | 3 Hours | ✓ | ✓ | $56.1 \pm 0.3$ | $37.8 \pm 10.7$ |
| Heuristic | N / A | | ✓ | ✓ | $51.0 \pm 0.0$ | N / A |
| 1-NN CV | $f_\theta(x), y$ | | ✓ | | $\mathbf{60.8 \pm 1.2}$ | $42.1 \pm 8.3$ |
| Logistic | $f_\theta(x), y$ | | ✓ | | $\mathbf{61.9 \pm 1.4}$ | $716.2 \pm 633.2$ |

Table 1: **Existing Work Underperforms on Diverse Source Models.** Average Pearson correlation and time taken per transfer averaged over the 6 target datasets of our benchmark for several existing model selection and transferability estimation methods. Some methods use source model probabilities $p_\theta(z \mid x)$, some use latent features $f_\theta(x)$, and some additionally require target data labels $y$. RSA and DDS need an extra model trained on the target data, for which we report average training time. All non-baseline methods either have almost no correlation (●) or take exorbitant amounts of time (●).

probabilities $p_\theta(z \mid x)$ over the source classes $z$, input target images $x$ and estimate a joint probability $\hat{p}(z, y)$ between the source classes $z$ and the target classes $y$. Then $\hat{p}_\theta(y \mid x)$ can be estimated as

$$\hat{p}_\theta(y \mid x) = \sum_{z_i} \hat{p}(y \mid z_i) \, p_\theta(z_i \mid x) \qquad (3)$$

Transferability is then computed by aggregating this probability distribution for all target images $x$. NCE and LEEP differ mainly in how they produce $\hat{p}(z, y)$: NCE (as extended in [34]) produces this by counting when the source model predicts $z$ for a target image with label $y$, while LEEP incorporates the entire distribution $p_\theta(z \mid x)$. The former naturally requires more data.

**Feature-Based Methods.** We also include several recent works from the Taskonomy [59] model selection literature: H-Score [4], RSA [11], and DDS [10]. All of these methods compare the penultimate layer's features $f_\theta(x)$ of a source model parameterized by $\theta$ evaluated on target images $x$ to a different set of features. H-score compares the covariance of the features with the covariance of their mean over the target classes $y$. Note that H-Score scales poorly with latent feature size, since they invert the covariance of the features (which can be as big as $4096 \times 4096$ for AlexNet [30]).

RSA [11] and DDS [10], on the other hand, compare the source model's extracted features to that of a "probe" model already trained on the target data. They assume that if two images are far apart in the probe model's feature space, then they should also be far apart in the feature space of a good source model. RSA and DDS vary in how they construct these distances and how they aggregate them to produce final scores: RSA uses one minus the correlation coefficient between each pair of images for distance and Spearman correlation for the final score, while DDS tests a large number of combinations, of which we choose the best performing in [10] (cosine distance and z-score). Note that both of these methods require training an additional model before they can begin recommending transfers, and the architecture used for this probe model highly impacts performance (see Tab. 1). Because these methods requires extra training time and expert knowledge of what architecture to use, we include them only for reference. While 3 extra hours of training time on a Titan Xp might not seem like much, it drastically limits accessibility on cheaper hardware, where training can take days.

**Baseline Methods.** Finally, we include three fairly standard baselines. The first two are simple classifiers learned on top of the penultimate layer's features $f_\theta(x)$ and tested on the probe set to get an estimate of final fine-tuning accuracy (outputting accuracy as the score). For these methods, we include $k = 1$ nearest neighbors with leave one out cross-validation (denoted as $k$-NN CV, used in [38]) and a logistic classifier trained on one half of the probe set and evaluated on the other half (used in [57]). We also include a simple heuristic that rates performance as the number of layers in the source network $\ell_s$ plus the log of the total number of images in both the source and target sets:

$$\text{heuristic}_s^t = \ell_s + \log\left(|\mathcal{D}^s| + |\mathcal{D}^t|\right) \qquad (4)$$

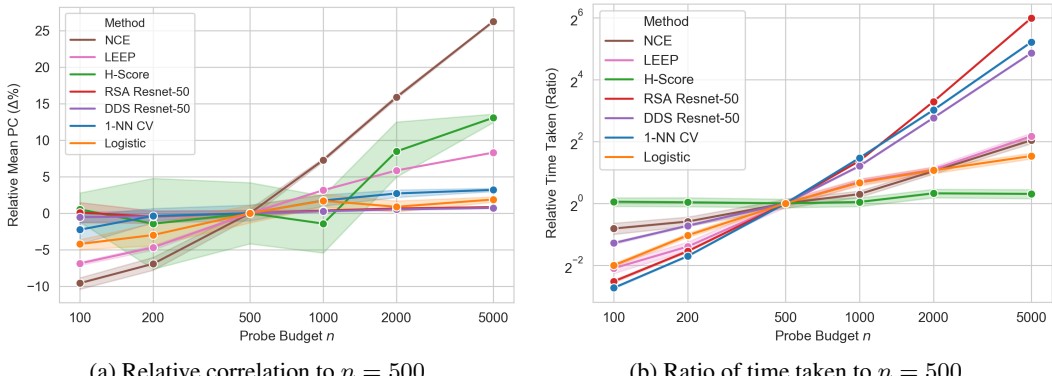

(a) Relative correlation to $n = 500$.  (b) Ratio of time taken to $n = 500$.

Figure 1: **Varying Probe Set Size.** We allow each model selection method to see a 500 image "probe set" of the target training data. To test whether this is a limiting factor for these methods, we vary probe set size and plot the resulting relative Pearson Correlation (a) and ratio of time taken (b) when compared to using 500 images. NCE [52], H-Score [4], and LEEP [34] all benefit significantly from more data (a), implying that their poor performance could be partially attributed to lack of data.

This captures the intuition that more layers and more data is better (with log data scaling).

**Results.** In Tab. 1 we list the Mean Pearson Correlation (computed as described in Sec. 3.1) and average time taken per transfer (not including feature extraction) for each method. We also include important properties of each method that can impact its scalability (training time, time taken) and whether it supports a diverse set of models (input requirements, being agnostic to source / target task).

From this experiment, we can see that current works fare poorly when applied to this new benchmark. In fact, none of these methods perform better than the simple $k = 1$ nearest neighbors baseline. Some even have close to no correlation (LEEP, NCE, H-Score). Furthermore, many of these methods are extremely expensive to compute (e.g., H-Score inverts a very large matrix when evaluating on AlexNet). RSA and DDS also require training of a target model with a manually selected architecture and high variability if poorly chosen (see Table 1). In the next section, we explore why these methods, which were not designed for our challenging transfer setting, may fail to generalize.

**A Note on Scalability.** This benchmark includes 168 transfers, which should be enough to test the scalability of model selection approaches. However, when evaluating current methods, we run the method on each source-target transfer and thus the "scalability" is linear per transfer. No current work exists to make this process sub-linear (e.g., through some kind of a weight embedding that you can binary search through), so instead we focus on the runtime speed of these methods. However, the end-goal of Scalable Diverse Model Selection is really to have sub-linear scaling, so we hope that future work can become even more scalable in that sense.

## 4  Analyzing Failure Modes of Existing Selection Methods

In this section, we explore what causes existing algorithms to fail in Sec. 3 and provide simple techniques to improve these methods on the challenging task of diverse model selection.

**Small Probe Set Size.** One potential reason why these methods fail to generalize to this setting is that, for efficiency, we use a probe set of $n = 500$ target train samples to compute transferability, while the final transfer model fine-tunes on all the data. In Fig. 1, we vary $n$ and show relative correlation (a) and the ratio of time taken (b) for each method. NCE, LEEP, and H-Score all gain a significant boost from the extra data, implying that perhaps the restriction of a probe set is a major limiting factor for their performance. On the other hand, RSA, DDS, and the baseline methods seem to be able to perform well with even fewer samples, which is a boon to scalability.

**Robustness to Evaluation Mode.** There's a subtle, but important, point about evaluation that's often overlooked in transferability estimation. In model selection, you assume that you have a fixed

| Method | Source Dataset & Arch | Architecture | Source Dataset | Target Dataset |
|---|---|---|---|---|
| NCE [52] | $2.1 \pm 0.7$ | $21.1 \pm 2.8$ | $5.8 \pm 0.4$ | $78.5 \pm 0.1$ |
| LEEP [34] | $10.8 \pm 0.1$ | $8.0 \pm 0.6$ | $20.7 \pm 0.2$ | $67.6 \pm 0.3$ |
| H-Score [4] | $-5.4 \pm 4.9$ | $-11.0 \pm 8.2$ | $3.1 \pm 8.8$ | $-51.7 \pm 7.0$ |
| RSA Resnet-50 [11] | $57.3 \pm 0.4$ | $\mathbf{67.5 \pm 0.6}$ | $61.5 \pm 0.3$ | $30.6 \pm 2.7$ |
| DDS Resnet-50 [10] | $56.1 \pm 0.3$ | $67.1 \pm 0.3$ | $62.3 \pm 0.3$ | $28.5 \pm 3.2$ |
| Heuristic | $51.05 \pm 0.0$ | $61.27 \pm 0.0$ | $7.13 \pm 0.0$ | $13.63 \pm 0.0$ |
| 1-NN CV | $\mathbf{60.8 \pm 1.2}$ | $\mathbf{67.4 \pm 2.0}$ | $67.0 \pm 0.9$ | $\mathbf{79.0 \pm 1.9}$ |
| Logistic | $\mathbf{61.9 \pm 1.4}$ | $\mathbf{68.7 \pm 2.9}$ | $\mathbf{68.3 \pm 1.5}$ | $\mathbf{81.0 \pm 1.7}$ |

Table 2: **Varying the Evaluation Mode.** In Eq. 2, we compute correlation over all source datasets and architectures. Here, we try varying each factor individually to more closely match each method's original setup (marked as ●). Most methods are poorly calibrated for other evaluation modes.

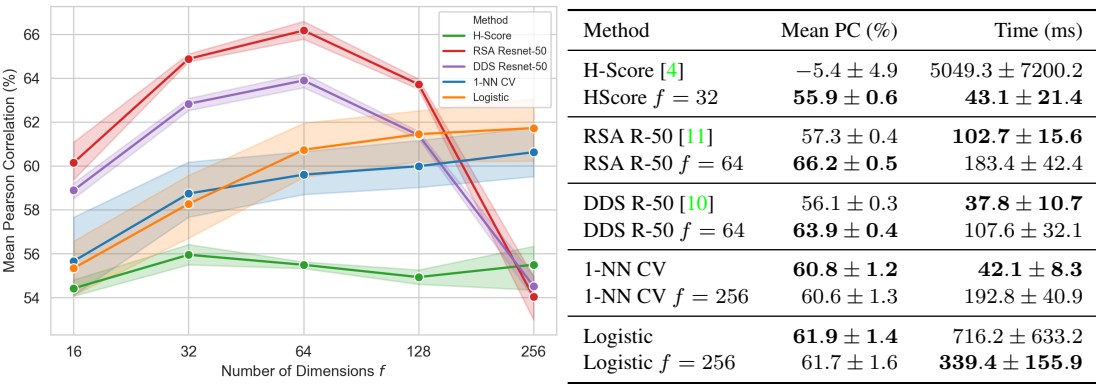

| Method | Mean PC (%) | Time (ms) |
|---|---|---|
| H-Score [4] | $-5.4 \pm 4.9$ | $5049.3 \pm 7200.2$ |
| HScore $f = 32$ | $\mathbf{55.9 \pm 0.6}$ | $\mathbf{43.1 \pm 21.4}$ |
| RSA R-50 [11] | $57.3 \pm 0.4$ | $\mathbf{102.7 \pm 15.6}$ |
| RSA R-50 $f = 64$ | $\mathbf{66.2 \pm 0.5}$ | $183.4 \pm 42.4$ |
| DDS R-50 [10] | $56.1 \pm 0.3$ | $\mathbf{37.8 \pm 10.7}$ |
| DDS R-50 $f = 64$ | $\mathbf{63.9 \pm 0.4}$ | $107.6 \pm 32.1$ |
| 1-NN CV | $\mathbf{60.8 \pm 1.2}$ | $\mathbf{42.1 \pm 8.3}$ |
| 1-NN CV $f = 256$ | $60.6 \pm 1.3$ | $192.8 \pm 40.9$ |
| Logistic | $\mathbf{61.9 \pm 1.4}$ | $716.2 \pm 633.2$ |
| Logistic $f = 256$ | $61.7 \pm 1.6$ | $\mathbf{339.4 \pm 155.9}$ |

(a) Varying the number of PCA components.

(b) PCA can improve performance and speed.

Figure 2: **Calibrating Features with PCA.** Feature-based methods with PCA applied to the features beforehand for different numbers of outputs dimensions $f$ (a). Applying feature reduction significantly improves all non-baseline methods (b), indicating that feature calibration was an issue. H-Score and Logistic, two methods that depend heavily on the number of features, also become much faster.

target dataset and would like to select the best source model for transfer. Transferability estimation often answers the opposite question: given a fixed source model, which target dataset will it best transfer to? The former evaluates following Eq. 2, while the latter considers the following correlation:

$$\text{Mean PC (Varying Target)} = \frac{1}{|S|} \sum_{s \in S} \text{pearsonr}(\{\alpha_s^t : t \in T\}, \{\omega_s^t : t \in T\}) \qquad (5)$$

One might assume that the same method would work well in both situations, but this is not the case.

In Tab. 2, we test different evaluation settings by swapping what set correlation is being computed over (i.e., $S$ for source and architecture, $T$ for target, and subsets of $S$ for the rest). For each non-baseline method, we've marked their native evaluation mode (using source dataset for Taskonomy). Most methods work well using their original evaluation protocol, but are not robust to other settings. Since each evaluation mode compares outputs across different factors, we hypothesize that the poor performance may be due to miscalibration and propose ways to mitigate this in the following sections.

**Dimensionality Reduction.** For the feature based methods, there's a lot of potential variation in the number of features between architectures. For instance, the number of features $|f_\theta(x)|$ for AlexNet is 4096, while for ResNet-18, it's 1024. Not only does this cause some methods to vary wildly in evaluation time (e.g., H-Score), but it also can be a source of miscalibration. To address this, we apply PCA dimensionality reduction [16] to the features, $f_\theta(x)$, down to a fixed dimension $f$ before computing each selection method. The results for this experiment are displayed in Fig. 2. All non-baseline feature-based methods receive a significant boost in performance from this dimensionality

| Method | Mean PC (%) Original | Mean PC (%) With Heuristic |
|---|---|---|
| LEEP [34] | $10.8 \pm 0.1$ | $\mathbf{26.8 \pm 0.1}$ |
| NCE [52] | $2.1 \pm 0.7$ | $\mathbf{28.3 \pm 0.5}$ |
| H-Score [4] | $-5.4 \pm 4.9$ | $\mathbf{25.6 \pm 9.2}$ |
| RSA R-50 [11] | $56.1 \pm 0.3$ | $\mathbf{65.7 \pm 0.2}$ |
| DDS R-50 [10] | $57.3 \pm 0.4$ | $\mathbf{64.9 \pm 0.2}$ |
| 1-NN CV | $60.8 \pm 1.2$ | $\mathbf{68.0 \pm 0.7}$ |
| Logistic | $61.9 \pm 1.4$ | $\mathbf{69.2 \pm 1.1}$ |

| Method | Mean PC (%) | Time (ms) |
|---|---|---|
| 1-NN CV $+\ell$ | $68.0 \pm 0.7$ | $42.1 \pm 8.3$ |
| Logistic $+\ell$ | $69.2 \pm 1.1$ | $716.2 \pm 633.2$ |
| RSA R-50 [11] | $57.3 \pm 0.4$ | $102.7 \pm 15.6$ |
| RSA R-50 $f = 32$ | $64.9 \pm 0.2$ | $183.4 \pm 42.4$ |
| RSA R-50 $+\ell, f = 32$ | $68.2 \pm 0.0$ | $183.4 \pm 42.4$ |
| **PARC** | $53.0 \pm 0.9$ | $49.4 \pm 18.3$ |
| **PARC** $f = 32$ | $59.3 \pm 0.7$ | $107.0 \pm 31.1$ |
| **PARC** $+\ell, f = 32$ | $\mathbf{70.3 \pm 0.5}$ | $107.0 \pm 31.1$ |

Table 3: **Modeling Capacity to Change.** In this experiment, we add the depth $\ell_s$ of the source model to each transferability prediction. This results in a significant performance boost for all methods on our benchmark.

Table 4: **PARC Results.** PARC compared with all the beneficial tweaks in Sec. 4 ($f$ for feature reduction and $+\ell$ for the layer heuristic, where $f = 32$ is best when combined). PARC outperforms all other methods, but requires these tweaks to work well.

reduction step. RSA and DDS even outperform the baselines after dimensionality reduction. For subsequent experiments, we will use the best value for $f$ found in Fig. 2a for each method.

**Capacity to Change.** The goal of a scalable diverse model selection algorithm is predict a score that correlates well with full fine-tuning on the target set. However, all methods so far have used the source model as fixed features or probabilities, without considering the potential for those features to change after fine-tuning. Thus, we observe poor performance on more difficult target datasets (such as NA Birds, see the Appendix), where models with a high capacity of learning are required. While modeling the capacity for a network to learn is out of the scope of this work, we can substitute that with a simple heuristic. A strong indicator of how much information a CNN can learn that applies to most off-the-shelf architectures (e.g., [30, 44, 48, 22, 50]) is simply the depth of the model (i.e., number of layers). In Tab. 3, we compare the performance on our benchmark for each method before and after adding this heuristic. To integrate this intuition, we first normalize the predicted scores $\alpha_s^t$ for each method by their mean $\mu^t$ and standard deviation $\sigma^t$ over all transfers and then add the relative source model depth $\ell_s$ over the maximum depth possible, $\ell_{\max}$ (i.e., $\ell_{\max} = 50$ for our experiments):

$$(\alpha_s^t)' = \frac{\alpha_s^t - \mu^t}{\sigma^t} + \frac{\ell_s}{\ell_{\max}} \tag{6}$$

This change results in a significant boost in performance for all methods and will be denoted as $+\ell$ for future experiments. We explore other ways to incorporate this model capacity in the Appendix.

## 5  Pairwise Annotation Representation Comparison (PARC)

Following the insights gained in Sec. 4, we devise a new method for the task of scalable diverse model selection. RSA [11] with dimensionality reduction performs extremely well (see Fig. 2), however it requires a "probe model" trained on the target data. To alleviate this restriction, we perform an intuitive modification: we replace the probe model with the ground truth labels. Features that work well for a target task should consider two images dissimilar if they were annotated differently.

More formally, given a probe set $\mathcal{P}_n$ of target images $x$ and labels $y$ and a model parameterized by $\theta$, PARC produces two distance matrices $D_\theta, D_y$ of shape $n \times n$ as

$$D_\theta = 1 - \text{corrcoef}(f_\theta(x)) \qquad D_y = 1 - \text{corrcoef}(g(y)) \tag{7}$$

where corrcoef computes pairwise Pearson product-moment correlation between the features of each pair of images (as used in RSA) and $g$ maps the labels $y$ to some vector representation. For classification, $g$ maps $y$ to the corresponding one-hot vector, but in general, $g$ can be any function that maps the annotations to a vector. For instance, with semantic segmentation as the target task, $g$ could produce a pixel-wise average of the annotations, and similar extensions exist for other computer vision tasks. We explore this further in Sec. 6.2.

Then, to compute the final PARC score, like RSA we simply compute the Spearman correlation between the two distance matrices for all pairs of images:

$$\text{PARC}(\theta, \mathcal{P}_n) = \text{spearmanr}(\{D_\theta[i,j] : i < j\}, \{D_y[i,j] : i < j\}) \tag{8}$$

| Method | Mean PC (%) | Time (ms) |
|---|---|---|
| RSA R-50 $+\ell$, $f = 64$ | $50.64 \pm 0.21$ | $180.6 \pm 29.3$ |
| DDS R-50 $+\ell$, $f = 64$ | $50.72 \pm 0.19$ | $120.4 \pm 26.2$ |
| 1-NN CV $+\ell$, $f = 256$ | $51.35 \pm 0.67$ | $209.6 \pm 61.5$ |
| **PARC** $+\ell$, $f = 32$ | $\mathbf{52.04 \pm 0.52}$ | $122.9 \pm 29.1$ |

Table 5: **Crowd-Sourced Models.** A more general benchmark obtained by training transfers from arbitrary crowd-sourced models. This includes 65 models from a variety of domains and 423 total transfers.

| Method | Training Time | PC (%) |
|---|---|---|
| RSA F-RCNN | 12 Hours | $\mathbf{96.33 \pm 0.31}$ |
| DDS F-RCNN | 12 Hours | $95.97 \pm 0.68$ |
| 1-NN CV | **None** | $89.67 \pm 6.09$ |
| **PARC** | **None** | $92.20 \pm 5.63$ |

Table 6: **Object Detection.** Predicting transfer performance for object detection. We employ a simple scheme to extend classification-based methods to object detection.

**Results.** We apply all the same improvements discussed in Sec. 4 and report the performance of PARC on our benchmark in Tab. 4. With both the dimensionality reduction and heuristic ensemble, PARC outperforms every other method (even with the same improvements). PARC observes a much larger boost from the heuristic ensemble than RSA, likely because RSA is allowed to use a ResNet-50 model fine-tuned on the target set. Thus, it already has some information about how features can change over fine-tuning built in (which is what the heuristic is trying to accomplish). PARC, on the other hand, isn't allowed this extra information, and thus adding the heuristic helps with calibration tremendously. We include more results for PARC in the Appendix.

## 6 Extended Benchmarks

Our benchmark in Sec. 3 describes a more practically useful setting than previous works, but it doesn't fully encapsulate scalable diverse model selection. Ideally, we could predict transferability between any source model (for any dataset, architecture, or task) to any target dataset or task. In this section, we explore arbitrary crowd-sourced source models and object detection as the target task.

### 6.1 A Crowd-Sourced Benchmark

In this experiment, we test how well PARC and other model selection methods perform on an even more diverse source model bank. To facilitate this, we collect 33 models from online model banks (i.e., [56, 18]) that span several source datasets ([7, 13, 32, 20, 51, 61, 33, 5]) and model architectures ([22, 42, 21, 27, 31]) covering many different pretext tasks (see the Appendix for full details). We combine this with the original 32 models we trained for a total of 65 source models and we add VOC2007 [13] multi-class classification as an additional target task, resulting in 423 transfers total. We place no restrictions on how the original models were trained, but we do normalize their features.

Results for this benchmark are available in Tab. 5. We only test on the subset of methods that support multi-class classification out of the box and apply all improvements from Sec. 4. While PARC outperforms the other methods, we note that none of the methods perform very well here (all having around 50% correlation with transfer accuracy). This indicates that selecting from an extremely diverse set of source models is still a very challenging task and warrants further study.

### 6.2 Transferability to Detection

So far, we've only considered classification as the target task. However, as mentioned in Sec. 5, we can apply PARC to other tasks by summarizing the annotations $y$ with a vector $g(y)$, e.g., by averaging the annotations pixel-wise. To average "pixel-wise" for object detection, we count all pixels belonging to boxes for each class, and then normalize by the total area of all boxes in the image. To extend 1-NN, we measure the $L_1$ distance between pairs of these aggregate label vectors.

In Tab. 6, we display the performance of each method for predicting transfers from 6 source detectors (Faster and Mask R-CNN [42, 21] on Cityscapes [5] and COCO [32], and Retinanet [31] and YOLOv3 [41] on just COCO) transferring to VOC2007 detection [13]. We compute Pearson correlation between the predicted scores and the mAP of each fine-tuned model, and we provide RSA and DDS a Faster R-CNN model trained on VOC2007. Because there are only 6 transfers, this is a much easier experiment than our main benchmark (and thus we found the techniques discussed in Sec. 4 to not

be important). While it doesn't perform as well as RSA or DDS here, PARC requires no additional model trained on the target data. Yet, it is still highly correlated with final fine-tuned mAP.

# 7 Conclusion and Limitations

In this work, we introduce Scalable Diverse Model Selection, create several benchmarks to test this task, analyze existing methods in this setting, address multiple techniques to improve performance, and finally iterate on existing methods to create a new approach that works well. While the techniques we found combined with PARC improve performance on this setting, there are a few limitations of our work. First, we assume that the model selection method is applied to every source model, which could get extremely expensive. Subsequent work could try relaxing this assumption. Second, lowering the barrier to entry to transfer learning makes deep learning more accessible, but the model returned by these algorithms isn't guaranteed to be the best, so more work would likely need to be done to temper the expectations of non-experts. Finally, selecting from any arbitrary set of models in a scalable way still remains challenging (especially on crowd-sourced models) and thus is still an open problem. Many factors can affect fine-tuned performance, but we only consider source feature quality and architecture capacity in this paper. We address this latter point with a simple heuristic, which is slightly unsatisfying. For instance, an ideal system could have some comprehensive learnability score for each architecture instead. Several papers look at factors that affect fine-tuning performance in isolation, but none combine everything into one recommendation system. We hope that this paper can be the first step in creating such a diverse model selection algorithm. We believe that a robust system to recommend pretrained models for transfer learning would be an incredible boon for the accessibility of deep learning and hope that future work can study this task in further detail.

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
