# Scalable Diverse Model Selection for Accessible Transfer Learning

## Supplemental Material (Appendix)

**Daniel Bolya**[*]
Georgia Tech
dbolya@gatech.edu

**Rohit Mittapalli**[*]
Georgia Tech
rmittapalli3@gatech.edu

**Judy Hoffman**
Georgia Tech
judy@gatech.edu

## Contents

---

[*]Equal Contribution

35th Conference on Neural Information Processing Systems (NeurIPS 2021).

# 1 Per-Target Results

There was not enough room in the main paper to display the Pearson correlation of each method for each target dataset. We display full results for all methods here.

Note that there are two datasets that typically stand out as being more difficult than the others to predict transfer accuracy for: NA Birds and CIFAR-10, both for opposite reasons. CIFAR-10 is the easiest target dataset available by far, where any model can perform well after fine-tuning. This means that the source feature quality doesn't matter nearly as much. Since source feature quality is the only metric these methods use to predict transfer performance, they do poorly here. On the other hand, NA Birds is challenging and thus requires a high capacity model to transfer well. These methods also fail to take this factor into account.

## 1.1 No Tweaks

In Tab. 1, we display the Pearson Correlation for each target dataset individually. We also include results for additional baselines and skews of existing methods. $k$-NN (without CV) is nearest neighbors that, instead of hold-one-out cross validation, trains on half the probe set and tests on the other half (like Logistic). RSA and DDS Full are where the probe model provided to RSA or DDS is the same architecture as the current model being queried. This doesn't do nearly as well, perhaps because the probe features need to be consistent across transfers. We also provided RSA and DDS with GoogLeNet and ResNet-18 models here. Neither of those do as well as their ResNet-50 counterpart, but they perform better than AlexNet. This reiterates the point that the architecture used for the RSA or DDS probe model must be the best one available.

| Method | Stan. Dogs | Ox. Pets | CUB 200 | NA Birds | CIFAR 10 | Caltech 101 | Mean PC |
|---|---|---|---|---|---|---|---|
| LEEP | $-6.3 \pm 0.2$ | $23.3 \pm 0.6$ | $25.8 \pm 0.2$ | $-5.9 \pm 0.2$ | $38.0 \pm 0.6$ | $-10.1 \pm 0.3$ | $10.8 \pm 0.1$ |
| NCE | $-8.8 \pm 0.2$ | $4.4 \pm 0.3$ | $13.3 \pm 0.6$ | $9.6 \pm 1.3$ | $15.3 \pm 2.4$ | $-21.2 \pm 0.5$ | $2.1 \pm 0.7$ |
| HScore | $-15.0 \pm 16.9$ | $-9.6 \pm 12.7$ | $9.5 \pm 22.6$ | $-3.8 \pm 29.7$ | $-5.3 \pm 10.8$ | $-8.4 \pm 5.6$ | $-5.4 \pm 4.9$ |
| 1-NN CV | $75.2 \pm 1.0$ | $71.2 \pm 1.7$ | $53.6 \pm 2.4$ | $50.9 \pm 4.2$ | $54.1 \pm 2.0$ | $59.6 \pm 0.9$ | $60.8 \pm 1.2$ |
| 5-NN CV | $75.0 \pm 0.9$ | $71.3 \pm 1.7$ | $53.0 \pm 3.4$ | $47.0 \pm 9.1$ | $51.7 \pm 1.1$ | $58.6 \pm 0.9$ | $59.4 \pm 1.9$ |
| 1-NN | $74.4 \pm 1.3$ | $70.7 \pm 2.7$ | $52.6 \pm 3.3$ | $51.9 \pm 5.0$ | $52.5 \pm 4.4$ | $59.8 \pm 1.6$ | $60.3 \pm 1.4$ |
| 5-NN | $75.4 \pm 1.9$ | $68.4 \pm 2.3$ | $58.1 \pm 5.8$ | $45.7 \pm 10.9$ | $47.6 \pm 5.4$ | $59.4 \pm 2.3$ | $59.1 \pm 3.5$ |
| Logistic | $75.2 \pm 0.8$ | $75.1 \pm 2.3$ | $55.6 \pm 3.4$ | $51.2 \pm 2.4$ | $51.5 \pm 3.3$ | $63.2 \pm 1.6$ | $61.9 \pm 1.4$ |
| RSA Resnet-50 | $63.3 \pm 0.6$ | $75.9 \pm 0.6$ | $50.5 \pm 1.5$ | $41.5 \pm 1.9$ | $48.7 \pm 1.5$ | $63.7 \pm 0.5$ | $57.3 \pm 0.4$ |
| RSA Resnet-18 | $63.0 \pm 0.9$ | $74.0 \pm 0.7$ | $37.7 \pm 1.8$ | $33.5 \pm 2.3$ | $50.8 \pm 1.7$ | $57.6 \pm 0.9$ | $52.8 \pm 0.7$ |
| RSA GoogLeNet | $47.8 \pm 0.7$ | $61.6 \pm 0.5$ | $5.9 \pm 2.2$ | $10.2 \pm 3.1$ | $31.5 \pm 1.9$ | $57.9 \pm 1.6$ | $35.8 \pm 0.6$ |
| RSA Alexnet | $-31.5 \pm 1.6$ | $27.9 \pm 1.3$ | $-17.0 \pm 1.5$ | $-45.6 \pm 0.7$ | $46.0 \pm 0.3$ | $11.8 \pm 1.1$ | $-1.4 \pm 0.6$ |
| RSA Full | $9.2 \pm 0.7$ | $63.0 \pm 0.6$ | $-32.1 \pm 0.9$ | $-67.8 \pm 0.8$ | $27.9 \pm 1.8$ | $31.0 \pm 0.9$ | $5.2 \pm 0.5$ |
| DDS Resnet-50 | $62.4 \pm 0.6$ | $75.4 \pm 0.5$ | $49.8 \pm 1.0$ | $38.5 \pm 0.9$ | $49.1 \pm 0.9$ | $61.2 \pm 0.5$ | $56.1 \pm 0.3$ |
| DDS Resnet-18 | $62.2 \pm 0.8$ | $74.2 \pm 0.5$ | $38.7 \pm 1.2$ | $31.8 \pm 1.2$ | $49.8 \pm 1.5$ | $57.4 \pm 0.4$ | $52.4 \pm 0.4$ |
| DDS GoogLeNet | $49.2 \pm 0.8$ | $64.6 \pm 0.6$ | $12.5 \pm 1.4$ | $12.6 \pm 1.8$ | $39.2 \pm 1.1$ | $57.7 \pm 1.1$ | $39.3 \pm 0.5$ |
| DDS Alexnet | $-27.8 \pm 1.1$ | $32.1 \pm 1.3$ | $-12.9 \pm 1.5$ | $-43.1 \pm 0.7$ | $47.2 \pm 1.0$ | $14.6 \pm 0.9$ | $1.7 \pm 0.4$ |
| DDS Full | $11.4 \pm 0.5$ | $63.5 \pm 0.7$ | $-27.0 \pm 0.6$ | $-63.3 \pm 0.8$ | $34.7 \pm 1.6$ | $30.9 \pm 0.9$ | $8.4 \pm 0.4$ |
| PARC | $58.9 \pm 1.1$ | $54.2 \pm 2.9$ | $45.0 \pm 0.4$ | $44.7 \pm 2.9$ | $47.0 \pm 1.1$ | $67.9 \pm 1.2$ | $53.0 \pm 0.9$ |

Table 1: **All Targets: No Tweaks.** Pearson Correlation for each target dataset without any tweaks applied for a budget size of $n = 500$.

## 1.2 Dimensionality Reduction

In Tab. 2, we show the Pearson Correlation for each dataset individually for each feature-based method while varying the dimensionality of the input features after PCA. The Mean PC column in this table is the data for the corresponding dimensionality reduction plot in the paper. We plot the means in Fig. 1.

| Method | Stan. Dogs | Ox. Pets | CUB 200 | NA Birds | CIFAR 10 | Caltech 101 | Mean PC |
|---|---|---|---|---|---|---|---|
| HScore $f = 16$ | $67.1 \pm 1.0$ | $68.2 \pm 0.6$ | $48.2 \pm 1.3$ | $32.4 \pm 2.2$ | $47.1 \pm 1.7$ | $63.6 \pm 0.8$ | $54.4 \pm 0.4$ |
| HScore $f = 32$ | $69.9 \pm 0.6$ | $67.8 \pm 0.8$ | $50.5 \pm 1.5$ | $35.2 \pm 0.8$ | $48.0 \pm 1.9$ | $64.3 \pm 0.5$ | $55.9 \pm 0.6$ |
| HScore $f = 64$ | $69.5 \pm 0.8$ | $69.2 \pm 1.0$ | $48.5 \pm 1.4$ | $31.0 \pm 1.2$ | $50.0 \pm 0.8$ | $64.8 \pm 0.5$ | $55.5 \pm 0.2$ |
| HScore $f = 128$ | $68.3 \pm 1.4$ | $70.3 \pm 1.0$ | $46.8 \pm 1.2$ | $27.3 \pm 2.6$ | $50.5 \pm 1.1$ | $66.4 \pm 0.6$ | $54.9 \pm 0.4$ |
| HScore $f = 256$ | $67.8 \pm 0.9$ | $71.2 \pm 1.4$ | $46.8 \pm 1.5$ | $30.0 \pm 3.4$ | $48.2 \pm 0.9$ | $69.0 \pm 0.6$ | $55.5 \pm 1.2$ |
| RSA R-50 $f = 16$ | $68.9 \pm 1.6$ | $73.2 \pm 0.6$ | $56.8 \pm 1.2$ | $45.4 \pm 3.5$ | $55.5 \pm 1.9$ | $61.2 \pm 1.4$ | $60.2 \pm 1.0$ |
| RSA R-50 $f = 32$ | $73.6 \pm 0.8$ | $71.7 \pm 0.4$ | $67.6 \pm 1.1$ | $52.6 \pm 1.1$ | $59.1 \pm 0.9$ | $64.8 \pm 0.5$ | $64.9 \pm 0.2$ |
| RSA R-50 $f = 64$ | $71.9 \pm 0.4$ | $67.1 \pm 0.6$ | $73.2 \pm 1.1$ | $63.4 \pm 2.3$ | $58.3 \pm 0.7$ | $63.2 \pm 0.6$ | $66.2 \pm 0.5$ |
| RSA R-50 $f = 128$ | $67.9 \pm 0.5$ | $61.2 \pm 0.3$ | $72.7 \pm 2.1$ | $65.1 \pm 1.4$ | $55.4 \pm 0.5$ | $60.0 \pm 0.3$ | $63.7 \pm 0.3$ |
| RSA R-50 $f = 256$ | $59.9 \pm 0.7$ | $54.9 \pm 0.6$ | $63.2 \pm 2.3$ | $41.1 \pm 3.2$ | $48.0 \pm 0.8$ | $57.1 \pm 0.7$ | $54.0 \pm 1.1$ |
| DDS R-50 $f = 16$ | $67.5 \pm 1.0$ | $71.7 \pm 0.5$ | $55.3 \pm 0.7$ | $42.7 \pm 1.6$ | $53.9 \pm 0.6$ | $62.2 \pm 1.5$ | $58.9 \pm 0.4$ |
| DDS R-50 $f = 32$ | $71.1 \pm 0.6$ | $72.2 \pm 0.5$ | $63.0 \pm 0.9$ | $47.3 \pm 1.4$ | $58.0 \pm 1.7$ | $65.4 \pm 0.5$ | $62.8 \pm 0.3$ |
| DDS R-50 $f = 64$ | $69.8 \pm 0.4$ | $71.2 \pm 0.5$ | $65.5 \pm 1.2$ | $51.9 \pm 1.2$ | $59.1 \pm 1.0$ | $65.8 \pm 0.4$ | $63.9 \pm 0.4$ |
| DDS R-50 $f = 128$ | $66.4 \pm 0.2$ | $66.7 \pm 0.3$ | $64.8 \pm 1.7$ | $47.6 \pm 0.4$ | $57.4 \pm 0.7$ | $65.6 \pm 0.5$ | $61.4 \pm 0.2$ |
| DDS R-50 $f = 256$ | $60.1 \pm 0.5$ | $58.9 \pm 0.6$ | $60.6 \pm 1.3$ | $34.7 \pm 1.6$ | $50.2 \pm 0.3$ | $62.7 \pm 0.6$ | $54.5 \pm 0.6$ |
| 1-NN CV $f = 16$ | $72.1 \pm 1.8$ | $70.1 \pm 1.9$ | $48.1 \pm 2.2$ | $35.3 \pm 6.1$ | $53.9 \pm 3.6$ | $54.6 \pm 1.5$ | $55.7 \pm 2.1$ |
| 1-NN CV $f = 32$ | $73.1 \pm 0.9$ | $71.0 \pm 1.5$ | $50.5 \pm 2.1$ | $45.9 \pm 5.8$ | $54.0 \pm 2.2$ | $58.0 \pm 1.8$ | $58.7 \pm 1.5$ |
| 1-NN CV $f = 64$ | $73.8 \pm 1.3$ | $72.0 \pm 1.9$ | $52.3 \pm 2.6$ | $46.8 \pm 4.8$ | $53.5 \pm 2.5$ | $59.2 \pm 0.9$ | $59.6 \pm 1.2$ |
| 1-NN CV $f = 128$ | $74.4 \pm 1.1$ | $71.7 \pm 1.8$ | $52.7 \pm 2.1$ | $48.2 \pm 4.0$ | $52.9 \pm 1.2$ | $59.9 \pm 0.6$ | $60.0 \pm 1.2$ |
| 1-NN CV $f = 256$ | $75.1 \pm 1.2$ | $71.1 \pm 1.6$ | $53.4 \pm 2.4$ | $50.1 \pm 4.8$ | $54.2 \pm 1.6$ | $59.9 \pm 0.8$ | $60.6 \pm 1.3$ |
| Logistic $f = 16$ | $72.0 \pm 1.7$ | $71.3 \pm 1.9$ | $46.1 \pm 3.1$ | $36.8 \pm 6.3$ | $49.1 \pm 6.0$ | $56.8 \pm 1.3$ | $55.3 \pm 1.5$ |
| Logistic $f = 32$ | $74.7 \pm 0.8$ | $72.2 \pm 2.7$ | $52.2 \pm 3.0$ | $43.0 \pm 4.5$ | $46.7 \pm 4.9$ | $60.8 \pm 0.9$ | $58.3 \pm 1.6$ |
| Logistic $f = 64$ | $75.3 \pm 1.2$ | $73.8 \pm 2.2$ | $53.6 \pm 2.9$ | $48.8 \pm 5.2$ | $50.0 \pm 4.3$ | $62.9 \pm 2.4$ | $60.7 \pm 1.4$ |
| Logistic $f = 128$ | $75.8 \pm 1.1$ | $74.1 \pm 2.9$ | $54.4 \pm 2.6$ | $49.2 \pm 3.5$ | $51.4 \pm 1.7$ | $63.7 \pm 1.2$ | $61.4 \pm 1.3$ |
| Logistic $f = 256$ | $76.1 \pm 0.7$ | $74.4 \pm 2.6$ | $55.0 \pm 2.8$ | $50.5 \pm 2.7$ | $50.8 \pm 2.8$ | $63.6 \pm 1.4$ | $61.7 \pm 1.6$ |
| PARC $f = 16$ | $64.3 \pm 2.2$ | $55.0 \pm 7.7$ | $44.7 \pm 1.5$ | $40.1 \pm 1.8$ | $49.9 \pm 2.3$ | $67.9 \pm 1.1$ | $53.6 \pm 1.1$ |
| PARC $f = 32$ | $68.6 \pm 0.6$ | $68.2 \pm 4.4$ | $49.8 \pm 3.0$ | $48.8 \pm 1.8$ | $50.5 \pm 1.9$ | $70.0 \pm 0.8$ | $59.3 \pm 0.7$ |
| PARC $f = 64$ | $71.4 \pm 0.9$ | $73.2 \pm 3.7$ | $53.3 \pm 3.0$ | $50.7 \pm 4.2$ | $50.1 \pm 1.4$ | $71.8 \pm 1.4$ | $61.7 \pm 1.3$ |
| PARC $f = 128$ | $72.8 \pm 1.5$ | $72.3 \pm 3.5$ | $57.0 \pm 2.1$ | $50.9 \pm 3.6$ | $46.7 \pm 2.0$ | $72.5 \pm 1.0$ | $62.0 \pm 0.7$ |
| PARC $f = 256$ | $75.3 \pm 1.0$ | $69.7 \pm 3.9$ | $60.1 \pm 2.3$ | $52.5 \pm 4.0$ | $38.8 \pm 2.6$ | $71.6 \pm 0.8$ | $61.3 \pm 0.6$ |

Table 2: **All Targets: Dimensionality Reduction.** Pearson Correlation for each target dataset with different levels of dimensionality reduction for a budget size of $n = 500$.

## 1.3 Capacity to Change

In Tab. 3, we include the Pearson Correlation for each dataset individually for all methods while incorporating the number of layers $\ell_s$ heuristic. We use this as a heuristic to gauge how well each source architecture can learn from complex data (with more layers predicting better performance). This helps tremendously on NA Birds, where a high capacity to learn is required to do well.

| Method | Stan. Dogs | Ox. Pets | CUB 200 | NA Birds | CIFAR 10 | Caltech 101 | Mean PC |
|---|---|---|---|---|---|---|---|
| LEEP | $14.6 \pm 0.2$ | $37.3 \pm 0.5$ | $46.3 \pm 0.1$ | $24.0 \pm 0.2$ | $29.6 \pm 0.4$ | $9.3 \pm 0.3$ | $26.8 \pm 0.1$ |
| NCE | $18.4 \pm 0.2$ | $20.9 \pm 0.2$ | $48.5 \pm 0.3$ | $51.9 \pm 1.4$ | $16.4 \pm 1.4$ | $13.4 \pm 0.3$ | $28.3 \pm 0.5$ |
| HScore | $24.5 \pm 27.9$ | $22.7 \pm 12.5$ | $40.6 \pm 27.7$ | $26.7 \pm 16.3$ | $2.4 \pm 9.5$ | $36.4 \pm 15.2$ | $25.6 \pm 9.2$ |
| 1-NN CV | $79.7 \pm 0.6$ | $70.6 \pm 1.5$ | $72.3 \pm 1.6$ | $75.5 \pm 1.6$ | $43.9 \pm 1.9$ | $66.0 \pm 0.6$ | $68.0 \pm 0.7$ |
| 5-NN CV | $79.5 \pm 0.5$ | $70.6 \pm 1.3$ | $74.9 \pm 2.1$ | $78.0 \pm 2.3$ | $42.1 \pm 0.7$ | $65.8 \pm 0.6$ | $68.5 \pm 0.7$ |
| 1-NN | $79.1 \pm 1.0$ | $70.1 \pm 2.4$ | $70.6 \pm 2.2$ | $73.6 \pm 2.4$ | $43.8 \pm 3.8$ | $65.6 \pm 1.1$ | $67.1 \pm 0.8$ |
| 5-NN | $79.8 \pm 1.5$ | $67.8 \pm 2.0$ | $77.3 \pm 1.0$ | $77.4 \pm 1.4$ | $40.0 \pm 4.4$ | $65.7 \pm 1.6$ | $68.0 \pm 1.2$ |
| Logistic | $80.2 \pm 0.7$ | $74.6 \pm 1.8$ | $74.0 \pm 2.4$ | $74.7 \pm 1.2$ | $41.3 \pm 2.9$ | $70.4 \pm 1.3$ | $69.2 \pm 1.1$ |
| RSA Resnet-50 | $70.8 \pm 0.5$ | $75.6 \pm 0.4$ | $67.7 \pm 0.9$ | $67.0 \pm 0.8$ | $44.2 \pm 1.5$ | $68.6 \pm 0.3$ | $65.7 \pm 0.2$ |
| RSA Resnet-18 | $72.3 \pm 0.7$ | $76.3 \pm 0.5$ | $69.6 \pm 1.0$ | $64.5 \pm 0.7$ | $44.4 \pm 1.5$ | $68.3 \pm 0.5$ | $65.9 \pm 0.4$ |
| RSA GoogLeNet | $58.2 \pm 0.5$ | $65.8 \pm 0.4$ | $33.6 \pm 1.8$ | $45.0 \pm 2.0$ | $30.0 \pm 1.6$ | $69.0 \pm 1.1$ | $50.3 \pm 0.4$ |
| RSA Alexnet | $-15.0 \pm 1.7$ | $41.3 \pm 1.5$ | $6.0 \pm 1.6$ | $-17.1 \pm 1.2$ | $46.8 \pm 0.5$ | $28.4 \pm 1.1$ | $15.1 \pm 0.6$ |
| RSA Full | $26.1 \pm 0.7$ | $73.3 \pm 0.5$ | $-7.4 \pm 0.8$ | $-42.0 \pm 1.1$ | $28.6 \pm 1.6$ | $50.8 \pm 0.8$ | $21.6 \pm 0.4$ |
| DDS Resnet-50 | $70.0 \pm 0.5$ | $75.4 \pm 0.4$ | $67.0 \pm 0.7$ | $63.9 \pm 0.4$ | $44.4 \pm 0.9$ | $68.9 \pm 0.3$ | $64.9 \pm 0.2$ |
| DDS Resnet-18 | $71.4 \pm 0.6$ | $76.1 \pm 0.4$ | $67.6 \pm 0.8$ | $62.0 \pm 0.4$ | $43.9 \pm 1.2$ | $69.0 \pm 0.3$ | $65.0 \pm 0.3$ |
| DDS GoogLeNet | $59.5 \pm 0.7$ | $68.5 \pm 0.4$ | $39.4 \pm 1.2$ | $46.3 \pm 1.2$ | $36.2 \pm 0.8$ | $69.1 \pm 0.8$ | $53.2 \pm 0.4$ |
| DDS Alexnet | $-11.0 \pm 1.1$ | $45.4 \pm 1.4$ | $10.7 \pm 1.4$ | $-13.9 \pm 1.2$ | $47.2 \pm 1.0$ | $32.0 \pm 0.7$ | $18.4 \pm 0.4$ |
| DDS Full | $28.0 \pm 0.5$ | $73.6 \pm 0.6$ | $-1.6 \pm 0.5$ | $-35.2 \pm 1.5$ | $34.7 \pm 1.5$ | $50.8 \pm 0.7$ | $25.0 \pm 0.4$ |
| PARC | $73.8 \pm 0.6$ | $57.5 \pm 2.0$ | $75.9 \pm 0.6$ | $79.2 \pm 0.8$ | $42.9 \pm 1.0$ | $74.4 \pm 0.7$ | $67.3 \pm 0.5$ |

Table 3: **All Targets: Capacity to Change.** Pearson Correlation for each target dataset with the $\ell_s$ heuristic incorporated for a budget size of $n = 500$.

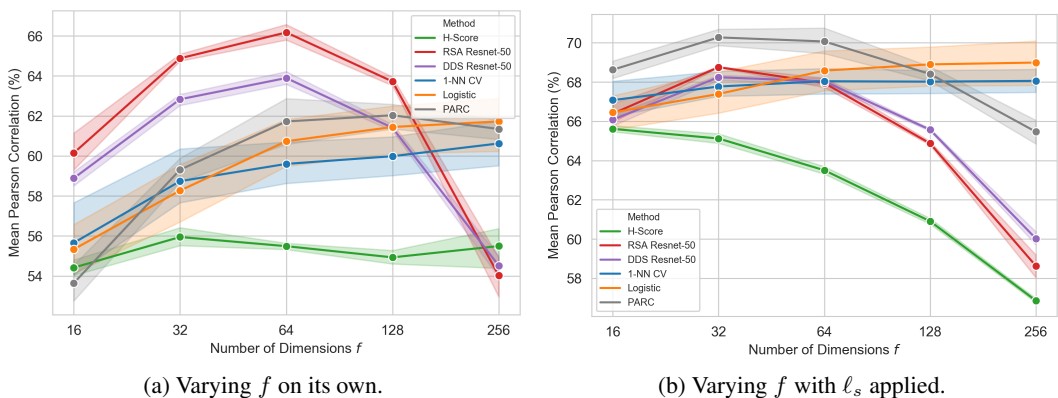

(a) Varying $f$ on its own.    (b) Varying $f$ with $\ell_s$ applied.

Figure 1: **Dimensionality Reduction.** In this figure, we vary the value of $f$ used for dimensionality reduction with and without the $\ell_s$ heuristic applied. The left plot is the same as the one in the main paper except with PARC added. Note that the best choice for $f$ is different with and without $\ell_s$. We failed to consider that in the original submission and will fix that for the final version.

## 1.4 All Tweaks

In Tab. 4, we show the Pearson Correlation for each dataset individually for all feature-based methods with both feature reduction and the $\ell_s$ heuristic applied. These are all tweaks outlined in our paper, and they improve the performance on all datasets significantly over each method's original versions. We plot the means in Fig. 1.

Note that the best dimensionality for feature reduction changes for most methods when the layer heuristic is applied. Thus, in our paper we report different values of $f$ in the final table than in the ablations for dimensionality reduction itself.

| Method | Stan. Dogs | Ox. Pets | CUB 200 | NA Birds | CIFAR 10 | Caltech 101 | Mean PC |
|---|---|---|---|---|---|---|---|
| HScore $f = 16$ | $77.0 \pm 0.6$ | $68.3 \pm 0.5$ | $74.6 \pm 0.9$ | $69.7 \pm 0.8$ | $31.4 \pm 0.9$ | $72.7 \pm 0.5$ | $65.6 \pm 0.2$ |
| HScore $f = 32$ | $78.1 \pm 0.5$ | $66.3 \pm 0.6$ | $76.8 \pm 0.5$ | $72.1 \pm 0.5$ | $24.7 \pm 0.4$ | $72.7 \pm 0.2$ | $65.1 \pm 0.3$ |
| HScore $f = 64$ | $76.6 \pm 0.4$ | $62.6 \pm 0.7$ | $77.6 \pm 0.4$ | $73.0 \pm 0.5$ | $18.7 \pm 0.1$ | $72.5 \pm 0.4$ | $63.5 \pm 0.2$ |
| HScore $f = 128$ | $74.2 \pm 0.5$ | $54.1 \pm 0.6$ | $77.8 \pm 0.2$ | $74.4 \pm 0.7$ | $14.1 \pm 0.1$ | $70.8 \pm 0.3$ | $60.9 \pm 0.2$ |
| HScore $f = 256$ | $70.0 \pm 0.3$ | $43.5 \pm 0.3$ | $76.7 \pm 0.3$ | $76.4 \pm 0.4$ | $11.4 \pm 0.0$ | $63.1 \pm 0.0$ | $56.9 \pm 0.1$ |
| RSA R-50 $f = 16$ | $74.9 \pm 1.2$ | $72.3 \pm 0.6$ | $72.5 \pm 0.8$ | $67.3 \pm 2.2$ | $44.6 \pm 0.8$ | $66.8 \pm 0.7$ | $66.4 \pm 0.6$ |
| RSA R-50 $f = 32$ | $77.2 \pm 0.8$ | $69.4 \pm 0.4$ | $78.0 \pm 0.7$ | $70.9 \pm 0.5$ | $48.8 \pm 0.7$ | $68.3 \pm 0.4$ | $68.8 \pm 0.0$ |
| RSA R-50 $f = 64$ | $74.6 \pm 0.4$ | $64.3 \pm 0.5$ | $79.7 \pm 0.6$ | $76.9 \pm 1.4$ | $46.7 \pm 0.5$ | $65.5 \pm 0.4$ | $67.9 \pm 0.3$ |
| RSA R-50 $f = 128$ | $70.3 \pm 0.4$ | $58.9 \pm 0.3$ | $77.9 \pm 1.2$ | $77.5 \pm 0.8$ | $43.0 \pm 0.4$ | $61.5 \pm 0.2$ | $64.9 \pm 0.1$ |
| RSA R-50 $f = 256$ | $63.2 \pm 0.5$ | $53.6 \pm 0.6$ | $72.6 \pm 1.1$ | $67.0 \pm 1.4$ | $36.2 \pm 0.7$ | $59.2 \pm 0.5$ | $58.6 \pm 0.6$ |
| DDS R-50 $f = 16$ | $74.1 \pm 0.8$ | $71.7 \pm 0.4$ | $71.7 \pm 0.5$ | $66.2 \pm 1.0$ | $44.6 \pm 0.5$ | $68.2 \pm 0.8$ | $66.1 \pm 0.3$ |
| DDS R-50 $f = 32$ | $76.0 \pm 0.5$ | $71.2 \pm 0.5$ | $76.2 \pm 0.6$ | $68.4 \pm 0.4$ | $47.7 \pm 1.3$ | $70.0 \pm 0.3$ | $68.2 \pm 0.2$ |
| DDS R-50 $f = 64$ | $73.9 \pm 0.3$ | $69.0 \pm 0.4$ | $77.4 \pm 0.6$ | $71.0 \pm 0.9$ | $47.1 \pm 0.7$ | $69.9 \pm 0.3$ | $68.0 \pm 0.2$ |
| DDS R-50 $f = 128$ | $70.1 \pm 0.2$ | $64.1 \pm 0.2$ | $76.7 \pm 1.0$ | $69.0 \pm 0.2$ | $44.6 \pm 0.5$ | $68.9 \pm 0.3$ | $65.6 \pm 0.1$ |
| DDS R-50 $f = 256$ | $64.1 \pm 0.4$ | $57.4 \pm 0.5$ | $73.2 \pm 0.6$ | $61.5 \pm 1.0$ | $38.4 \pm 0.3$ | $65.6 \pm 0.3$ | $60.0 \pm 0.4$ |
| 1-NN CV $f = 16$ | $78.5 \pm 1.2$ | $70.9 \pm 1.5$ | $71.3 \pm 1.4$ | $72.0 \pm 2.4$ | $44.3 \pm 2.5$ | $65.4 \pm 0.9$ | $67.1 \pm 1.0$ |
| 1-NN CV $f = 32$ | $78.7 \pm 0.4$ | $71.2 \pm 1.2$ | $71.4 \pm 1.3$ | $74.4 \pm 2.3$ | $44.5 \pm 2.0$ | $66.4 \pm 1.1$ | $67.8 \pm 0.8$ |
| 1-NN CV $f = 64$ | $79.1 \pm 0.7$ | $71.7 \pm 1.6$ | $72.2 \pm 1.5$ | $74.6 \pm 2.1$ | $43.7 \pm 2.1$ | $67.0 \pm 0.7$ | $68.0 \pm 0.7$ |
| 1-NN CV $f = 128$ | $79.3 \pm 0.7$ | $71.4 \pm 1.4$ | $72.2 \pm 1.4$ | $74.9 \pm 1.7$ | $43.2 \pm 1.2$ | $67.1 \pm 0.5$ | $68.0 \pm 0.7$ |
| 1-NN CV $f = 256$ | $79.7 \pm 0.6$ | $70.6 \pm 1.3$ | $72.2 \pm 1.7$ | $75.3 \pm 1.7$ | $44.0 \pm 1.5$ | $66.6 \pm 0.5$ | $68.1 \pm 0.7$ |
| Logistic $f = 16$ | $78.8 \pm 1.2$ | $71.5 \pm 1.3$ | $70.6 \pm 1.4$ | $70.9 \pm 2.5$ | $39.8 \pm 4.3$ | $67.1 \pm 0.7$ | $66.5 \pm 1.0$ |
| Logistic $f = 32$ | $80.3 \pm 0.5$ | $72.3 \pm 2.1$ | $72.4 \pm 1.6$ | $73.0 \pm 2.9$ | $37.6 \pm 3.8$ | $68.7 \pm 0.3$ | $67.4 \pm 1.3$ |
| Logistic $f = 64$ | $80.4 \pm 1.0$ | $73.1 \pm 1.5$ | $73.1 \pm 1.4$ | $74.8 \pm 2.6$ | $40.1 \pm 3.1$ | $70.0 \pm 2.2$ | $68.6 \pm 1.2$ |
| Logistic $f = 128$ | $80.6 \pm 1.0$ | $73.1 \pm 2.1$ | $73.3 \pm 1.6$ | $74.4 \pm 1.9$ | $41.2 \pm 2.0$ | $70.7 \pm 1.0$ | $68.9 \pm 1.1$ |
| Logistic $f = 256$ | $80.8 \pm 0.8$ | $73.4 \pm 2.1$ | $73.8 \pm 1.8$ | $74.6 \pm 1.9$ | $40.8 \pm 2.5$ | $70.6 \pm 1.1$ | $69.0 \pm 1.3$ |
| PARC $f = 16$ | $76.9 \pm 1.2$ | $61.1 \pm 3.6$ | $75.8 \pm 0.9$ | $76.4 \pm 0.7$ | $45.1 \pm 1.9$ | $76.6 \pm 1.0$ | $68.6 \pm 0.5$ |
| PARC $f = 32$ | $78.3 \pm 0.4$ | $66.9 \pm 2.2$ | $76.3 \pm 1.4$ | $77.2 \pm 0.8$ | $45.1 \pm 1.6$ | $78.0 \pm 0.5$ | $70.3 \pm 0.5$ |
| PARC $f = 64$ | $78.4 \pm 0.7$ | $67.2 \pm 2.2$ | $76.1 \pm 1.3$ | $76.5 \pm 1.8$ | $43.6 \pm 1.2$ | $78.7 \pm 1.0$ | $70.1 \pm 0.8$ |
| PARC $f = 128$ | $78.1 \pm 1.2$ | $64.8 \pm 2.3$ | $76.6 \pm 0.7$ | $74.6 \pm 2.0$ | $38.5 \pm 1.6$ | $77.9 \pm 0.7$ | $68.4 \pm 0.4$ |
| PARC $f = 256$ | $79.0 \pm 0.9$ | $59.6 \pm 2.8$ | $76.1 \pm 1.5$ | $72.5 \pm 2.4$ | $30.3 \pm 1.4$ | $75.4 \pm 1.1$ | $65.5 \pm 0.7$ |

Table 4: **All Targets: All Tweaks.** Pearson Correlation for each target dataset for each feature-based method with PCA reduction down to $f$ features and with the $\ell_s$ heuristic incorporated for a budget size of $n = 500$.

## 2    More Results for Varying Probe Set Size

In the main paper, we only varied the probe set size for the original version of each method. Because the original methods all had vastly different correlations and time taken, we had to plot relative correlation and the ratio of time taken instead. This showed how different methods scaled differently with extra data, but it didn't show how well each method performed relative to each other. In Fig. 2, we show the performance and time taken while varying the probe set size $n$ for each competitive method with all beneficial tweaks applied. These plots are absolute so that the performance of each method can be compared.

PARC outperforms all other methods for all values of $n$ in this setting, though using more data requires exponentially more time. This is mostly because PARC computes correlations on an $n \times n$ matrix, meaning it scales poorly with $n$. Logistic scales better in time taken than PARC, becoming faster at $n = 5000$. While it's still slightly worse than PARC, it's a good alternative to use if you plan to use the full dataset, for instance.

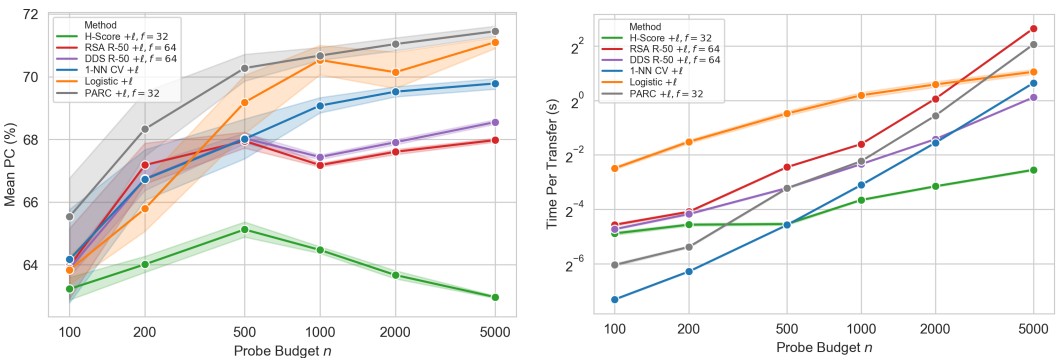

(a) Mean PC of each method while varying $n$.    (b) Time taken by each method while varying $n$.

Figure 2: **Varying Probe Set Size.** In this figure, we vary the probe set size $n$ for the best version of each competitive method (with all useful tweaks applied). Note that the absolute performance is plotted here, not relative like in the main paper.

## 3 Other Ways to Model Capacity to Change

In Section 4 of the main paper, we ensemble a heuristic with each method according to the following equation:

$$(\alpha_s^t)'_1 = \frac{\alpha_s^t - \mu^t}{\sigma^t} + \frac{\ell_s}{\ell_{\max}} \tag{1}$$

This takes the form of normalizing the method's output scores and then adding the ratio between the number of layers in the network $\ell_s$ and the number of layers in the deepest network $\ell_{\max}$. For all experiments, we used $\ell_{\max} = 50$, because the deepest network we used was ResNet-50.

In this section, we test more ways to perform this heuristic ensemble. Since all approaches output a different range of scores, it was important to first normalize the output scores to be consistent across methods. In the original paper, we subtract by the mean and divide by the standard deviation over all source models to accomplish this normalization. However, another way to normalize the scores would be to re-scale the empirical min to 0 and empirical max to 1.

$$(\alpha_s^t)'_2 = \frac{\alpha_s^t - \min^t}{\max^t - \min^t} + \frac{\ell_s}{\ell_{\max}} \tag{2}$$

This is a more intuitive form of normalization, but it doesn't perform as well as Eq. 1 because it puts too much weight on the number of layers. To address this, we can simply scale the heuristic term:

$$(\alpha_s^t)'_3 = \frac{\alpha_s^t - \min^t}{\max^t - \min^t} + \lambda_\ell \frac{\ell_s}{\ell_{\max}} \tag{3}$$

We find $\lambda_\ell = \frac{1}{4}$ to be a good choice. In Tab. 5, we benchmark the performance of these ensembling equations for each method. Eq. 1 works the best for all methods (within error bars), with Eq. 3 working similarly well for around half the methods. Note that Eq. 2 works the best only for LEEP and NCE, indicating that their distribution of outputs is very different to that of the other methods (which makes sense given that they both output a log-based score). We could find a value for $\lambda_\ell$ that maximizes the performance of each method, but this risks overfitting to our benchmark. We leave hyperparemeter tuning on a separate benchmark to future work. Thus, because it requires no hyperparameter tuning and works on a broad range of methods, we use Eq. 1 in our original paper.

| Method | Mean PC (%) Original | Mean PC (%) Eq. 1 | Mean PC (%) Eq. 2 | Mean PC (%) Eq. 3 |
|---|---|---|---|---|
| LEEP | $10.82 \pm 0.13$ | $26.83 \pm 0.12$ | $\mathbf{47.05 \pm 0.09}$ | $26.03 \pm 0.12$ |
| NCE | $2.08 \pm 0.67$ | $28.27 \pm 0.47$ | $\mathbf{48.15 \pm 0.14}$ | $29.75 \pm 0.45$ |
| H-Score $f = 32$ | $55.95 \pm 0.55$ | $\mathbf{65.12 \pm 0.28}$ | $60.44 \pm 0.12$ | $\mathbf{65.08 \pm 0.29}$ |
| RSA R-50 $f = 64$ | $66.17 \pm 0.47$ | $\mathbf{67.95 \pm 0.28}$ | $61.24 \pm 0.09$ | $67.42 \pm 0.23$ |
| DDS R-50 $f = 64$ | $63.90 \pm 0.37$ | $\mathbf{68.04 \pm 0.20}$ | $62.80 \pm 0.05$ | $\mathbf{68.08 \pm 0.17}$ |
| 1-NN CV | $60.75 \pm 1.25$ | $\mathbf{68.01 \pm 0.75}$ | $63.95 \pm 0.41$ | $\mathbf{67.95 \pm 0.76}$ |
| Logistic | $61.94 \pm 1.42$ | $\mathbf{69.18 \pm 1.10}$ | $64.60 \pm 0.53$ | $\mathbf{69.11 \pm 1.13}$ |
| PARC $f = 32$ | $59.31 \pm 0.68$ | $\mathbf{70.28 \pm 0.47}$ | $60.80 \pm 0.19$ | $69.75 \pm 0.42$ |

Table 5: **Adding Capacity to Change.** Different ways to incorporate the number of layers heuristic $\ell_s/\ell_{\max}$. The best ensembling methods (and those within one standard deviation of the best) for each method are shown in bold for each method. For all competetive methods, Eq. 1 works the best, so that is what we use in the original paper.

# 4    Sources for the Crowd-Sourced Benchmark

In Tab. 6 we list all source models use in our crowd-sourced benchmark (Sec. 6.2 in the original paper). To extract features from each ResNet-based architecture, we simply globally pool the features in the C5 layer. Also, while the crowd-sourced models contain ResNet-101 with $\ell_s = 101$, we still fix $\ell_{max} = 50$ for Eq. 1 to be consistent with the original benchmark.

| Method | Backbone | Dataset | Model ID | License | Source |
|--------|----------|---------|----------|---------|--------|
| Faster R-CNN | ResNet-101 C4 | COCO | 138204752 | CC BY-SA 3.0 | Detectron v2 |
| Faster R-CNN | ResNet-50 C4 | COCO | 137257644 | CC BY-SA 3.0 | Detectron v2 |
| Faster R-CNN | ResNet-50 C4 | COCO | 137849393 | CC BY-SA 3.0 | Detectron v2 |
| Faster R-CNN | ResNet-50 C4 | VOC 07+12 | 142202221 | CC BY-SA 3.0 | Detectron v2 |
| Faster R-CNN | ResNet-50 FPN | COCO | 137257794 | CC BY-SA 3.0 | Detectron v2 |
| Faster R-CNN | ResNet-50 FPN | COCO | 137849458 | CC BY-SA 3.0 | Detectron v2 |
| Faster R-CNN | ResNet-101 FPN | COCO | 137851257 | CC BY-SA 3.0 | Detectron v2 |
| Mask R-CNN | ResNet-101 FPN | COCO | 138205316 | CC BY-SA 3.0 | Detectron v2 |
| Mask R-CNN | ResNet-101 C4 | COCO | 138363239 | CC BY-SA 3.0 | Detectron v2 |
| Mask R-CNN | ResNet-50 C4 | COCO | 137259246 | CC BY-SA 3.0 | Detectron v2 |
| Mask R-CNN | ResNet-50 C4 | COCO | 137849525 | CC BY-SA 3.0 | Detectron v2 |
| Mask R-CNN | ResNet-50 FPN | COCO | 137260431 | CC BY-SA 3.0 | Detectron v2 |
| Mask R-CNN | ResNet-50 FPN | COCO | 137849600 | CC BY-SA 3.0 | Detectron v2 |
| Mask R-CNN | ResNet-50 FPN | Cityscapes | 142423278 | CC BY-SA 3.0 | Detectron v2 |
| Mask R-CNN | ResNet-50 FPN | LVIS | 144219072 | CC BY-SA 3.0 | Detectron v2 |
| Mask R-CNN | ResNet-101 FPN | LVIS | 144219035 | CC BY-SA 3.0 | Detectron v2 |
| Keypoint R-CNN | ResNet-101 FPN | COCO | 138363331 | CC BY-SA 3.0 | Detectron v2 |
| Keypoint R-CNN | ResNet-50 FPN | COCO | 137261548 | CC BY-SA 3.0 | Detectron v2 |
| Keypoint R-CNN | ResNet-50 FPN | COCO | 137849621 | CC BY-SA 3.0 | Detectron v2 |
| Panoptic R-CNN | ResNet-101 FPN | COCO | 139514519 | CC BY-SA 3.0 | Detectron v2 |
| Panoptic R-CNN | ResNet-101 FPN | COCO | 139797668 | CC BY-SA 3.0 | Detectron v2 |
| Panoptic R-CNN | ResNet-50 FPN | COCO | 139514544 | CC BY-SA 3.0 | Detectron v2 |
| Panoptic R-CNN | ResNet-50 FPN | COCO | 139514569 | CC BY-SA 3.0 | Detectron v2 |
| RetinaNet | ResNet-101 | COCO | 190397697 | CC BY-SA 3.0 | Detectron v2 |
| RetinaNet | ResNet-50 | COCO | 190397773 | CC BY-SA 3.0 | Detectron v2 |
| RetinaNet | ResNet-50 | COCO | 190397829 | CC BY-SA 3.0 | Detectron v2 |
| SimCLR | ResNet-101 | ImageNet 1k | | MIT | VISSL |
| ClusterFit | ResNet-50 | ImageNet 1k | | MIT | VISSL |
| DeepCluster v2 | ResNet-50 | ImageNet 1k | | MIT | VISSL |
| Jigsaw | ResNet-50 | ImageNet 22k | | MIT | VISSL |
| MOCO | ResNet-50 | ImageNet 1k | | MIT | VISSL |
| NPID | ResNet-50 | ImageNet 1k | | MIT | VISSL |
| PIRL | ResNet-50 | ImageNet 1k | | MIT | VISSL |
| RotNet | ResNet-50 | ImageNet 22k | | MIT | VISSL |
| SimCLR | ResNet-50 | ImageNet 1k | | MIT | VISSL |
| SWAV | ResNet-50 | ImageNet 1k | | MIT | VISSL |
| Semi-Supervised | ResNet-50 | Instagram | | MIT | VISSL |
| Semi-Supervised | ResNet-50 | YFCC100M | | MIT | VISSL |
| Supervised | ResNet-50 | Places205 | | MIT | VISSL |

Table 6: **Crowd-Sourced Source Models.** For our crowd-sourced benchmark, we download pre-trained source models from two model banks. For each source model, we list the originating method, the architecture it uses, the dataset it was trained on, the corresponding model ID, the license the source model was released under, and the source model bank. For models without an ID (i.e., VISSL models), we take the highest performing model released.

# 5 A Note on Metrics

**Is Pearson Correlation a suitable metric.** We believe that Pearson Correlation (instead of some top-k accuracy) is the right metric to use for Scalable Diverse Model Selection for several reasons:

1. We want our results to be meaningful no matter the model bank being used. If we just looked at the top model output, our results would be entirely invalidated if models were added or removed to the model bank. We care about the intrinsic quality of the selection models themselves, rather than specifically how they perform on our exact model bank.

2. Our goal is to address diverse model selection, meaning that the source models should vary in architecture, dataset, and task. If we only look at the top model returned by the selection algorithm, we're not evaluating how well that method can compare across these source model variations. In order to do that, we need to take the ranking of all the models into account.

3. Practitioners using these model selection algorithms will have different needs depending on their situation. If they need to run their models on phones or other low-power devices, ResNet-50 might not be an option. They'd want to compare e.g., ResNet-18, GoogLeNet, and MobileNet among other more optimized architectures, meaning our evaluation method must take that into account.

**Other Metrics.** Nevertheless, it would be useful to have additional metrics that depend more on the actual top results predicted by the method. For this purpose, we think relative accuracy would be the easiest to interpret. That is, take the top-k models suggested by the algorithm, fetch their transfer performance and average those numbers. Finally, divide this average by the transfer performance of the best model to obtain a notion of "model selection accuracy". Here are the results for different values of k. We list columns with the original performance of each method (w/o the tricks discussed in the paper), as well as the same with the tricks added (w/ Tricks).

| Method | Original, $k = 1$ | Original, $k = 3$ | w/ Tricks, $k = 1$ | w/ Tricks, $k = 3$ | w/ Tricks, $k = 5$ |
|---|---|---|---|---|---|
| LEEP | $93.18\% \pm 0.00$ | $90.72\% \pm 0.04$ | $99.56\% \pm 0.00$ | $92.79\% \pm 0.02$ | $90.86\% \pm 0.00$ |
| NCE | $95.80\% \pm 1.65$ | $89.66\% \pm 1.62$ | $98.82\% \pm 0.40$ | $96.80\% \pm 0.26$ | $94.56\% \pm 0.25$ |
| HScore | $84.78\% \pm 2.77$ | $83.55\% \pm 2.17$ | $99.46\% \pm 0.10$ | $98.63\% \pm 0.04$ | $97.66\% \pm 0.14$ |
| RSA R-50 | $98.27\% \pm 0.12$ | $98.32\% \pm 0.15$ | $99.35\% \pm 0.04$ | $98.66\% \pm 0.03$ | $97.75\% \pm 0.11$ |
| DDS R-50 | $99.37\% \pm 0.00$ | $98.25\% \pm 0.17$ | $99.37\% \pm 0.01$ | $98.62\% \pm 0.03$ | $97.70\% \pm 0.09$ |
| 1-NN CV | $99.60\% \pm 0.05$ | $97.07\% \pm 0.79$ | $99.60\% \pm 0.05$ | $97.75\% \pm 0.27$ | $96.94\% \pm 0.02$ |
| Logistic | $99.58\% \pm 0.03$ | $96.46\% \pm 0.92$ | $99.45\% \pm 0.17$ | $97.86\% \pm 0.26$ | $97.08\% \pm 0.27$ |
| PARC | · | · | $99.31\% \pm 0.18$ | $98.43\% \pm 0.12$ | $97.88\% \pm 0.00$ |

Table 7: **Top-K Relative Accuracy.** The accuracy of the predicted model divided by the accuracy of the best possible model averaged across the top $k$ models for each modle selection method. This only considers the top models, but this gives an estimate of how well each method does in terms of comparing raw accuracy.

Three points are clear from these results:

1. Adding the tricks described in the paper significantly improves the top-k results for almost all methods. This seems to be true no matter what mode of analysis we use.

2. With the tricks applied, some methods like LEEP have very high top-1 accuracies, but quickly fall off as more than 1 model is taken into account. Top-k results like these are inherently flawed in that they depend significantly on the list of models used, as small changes in that list can quickly shake up which algorithm performs the best, which is alleviated by a metric like Pearson Correlation which takes all models into account.

3. While PARC w/ Tricks isn't explicitly at the top except for $k \geq 5$, it's still a very strong contender that's fast and can be applied to any source task, architecture, or target task. Moreover, it's well calibrated for all types of models (which is what the Pearson Correlation results show).

We'd like to reiterate that this metric only tests 1-5 of the top selected models and does not test whether the algorithm is robust across source, architecture, or task (as those variations aren't likely to come up in the top 5 models). Thus, we keep Pearson Correlation as our main metric for evaluation, since top-k metrics don't capture our goal of testing a method's robustness to model diversity.