# OpenReview forum: "Scalable Diverse Model Selection for Accessible Transfer Learning"
_NeurIPS.cc/2021/Conference — NeurIPS 2021 Poster_

### Official Review · Reviewer_LikY · 2021-07-12

**Rating:** 5
**Confidence:** 5

**Summary:**

This paper introduced a unified benchmark to test the performance of existing transferability estimation algorithms. The main contribution is firstly its unified benchmark, and then, its experiment models and datasets are large, which show a fair comparison for all methods. Finally, they modified an existing algorithm to propose a new one, which achieves better Person correlation score over others.

**Limitations And Societal Impact:**

No negative societal impact

**Main Review:**

1. My main concern is the applicability of the transferability estimation algorithms in real applications. From the recent results of LEEP, NCE, and the recent ones (LogME [1] and OTCE [2]), we see that although their evaluation scores (e.g., Person correlation) vary, but the results seem quite unified: no matter what algorithm you use, the most suitable source model will always be the biggest one (say, all methods will select ResNet-50 over ResNet-18 since we know that ResNet-50 has more capacity). So I question this line of research by the following questions:
- Is it suitable to use a simple Person correlation score as the evaluation metric? Selecting the best model is a 0/1 task, if all methods give the correct model, then it is not necessary to measure their scores.
- If all methods select the model with biggest capacity  / biggest pre-trained dataset, then what's the role of these selection algorithms? We can simply use the big models as the source model, why bother selecting?
- Last, the Person correlation score is not that convincing: e.g. it is hard to understand what an improvement of 3% in Person correlation stands for.

2. It is hard to understand the word "scalable". I think all other methods can also work in the situation of increasing source models / target models. Plus, from the results, the time complexity of PARC is just comparable to others, making me really confusing about this claim.

3. There is no theoretical justification why this selection method can work.

[1] You K, Liu Y, Wang J, et al. Logme: Practical assessment of pre-trained models for transfer learning[C]//International Conference on Machine Learning. PMLR, 2021: 12133-12143.

[2] Tan Y, Li Y, Huang S L. OTCE: A Transferability Metric for Cross-Domain Cross-Task Representations[C]//Proceedings of the IEEE/CVF Conference on Computer Vision and Pattern Recognition. 2021: 15779-15788.

--- POST REBUTTAL ---
Author's rebuttal did address some of my concerns. But I still think this paper has some issues that need to be addressed. Plus, the writing  should also be improved. Thus, I keep my score.

**Time Spent Reviewing:**

3

---

> ### Author Response · Authors · 2021-08-10
> **Author Response to Reviewer LikY**
>
> Thank you for your comments and for pointing us to LogME and OTCE as more related work! We'll include those papers in the paper discussion. Then, we definitely hear your concerns and were also wondering whether the biggest model trained with the most data would just be the best choice in all settings. We included a heuristic that ranks the models by architecture and dataset size in the paper for this very reason and found it to work worse than the best methods (see Tab. 1 with the heuristic defined in Eq. 4). Though, that's evaluated using Pearson Correlation (PC), meaning it takes all the source models into account.
>
> **Is Pearson Correlation a suitable metric?** We believe that Pearson Correlation (instead of some top-k accuracy) is the right metric to use for this task for several reasons:
>  1. We want our results to be meaningful no matter the model bank being used. If we just looked at the top model output, our results would be entirely invalidated if models were added or removed to the model bank. We care about the intrinsic quality of the selection models themselves, rather than specifically how they perform on our exact model bank.
>  2. Our goal is to address *diverse* model selection, meaning that the source models should vary in architecture, dataset, and task. If we only look at the top model returned by the selection algorithm, we're not evaluating how well that method can compare across these source model variations. In order to do that, we need to take the ranking of all the models into account.
>  3. Practitioners using these model selection algorithms will have different needs depending on their situation. If they need to run their models on phones or other low-power devices, ResNet-50 might not be an option. They'd want to compare e.g., ResNet-18, GoogLeNet, and MobileNet among other more optimized architectures, meaning our evaluation method must take that into account.
>
> **Is the biggest model always better?** Nevertheless, you are right in that LEEP and NCE's top predictions are often worse than the biggest model (ResNet-50 w/ ImageNet). However, the biggest model is not always the best for every target dataset. Using the methods introduced in this paper, we can actually select a better model than ResNet-50 w/ ImageNet on average.
>
> In the following table, we report the average absolute *difference in accuracy* of the top model selected vs. ResNet-50 w/ ImageNet (i.e., [Top Model Acc] - [R-50 ImageNet Acc]). Positive means better than the largest model on average, while negative means worse.
>
>
> | Method    | Original            | With PCA + Heuristic | With PCA + Heuristic (only GoogLeNet) |
> |:----------|:-------------------:|:--------------------:|:-------------------------------------:|
> | LEEP      |  -4.12% +/- 0.00    | 0.80% +/- 0.00       | 2.98% +/- 0.26                        |
> | NCE       |  -2.19% +/- 1.26    | 0.17% +/- 0.34       | 0.86% +/- 1.40                        |
> | HScore    | -11.72% +/- 2.17    | 0.73% +/- 0.08       | 4.22% +/- 0.05                        |
> | RSA R-50  |  -0.28% +/- 0.10    | 0.49% +/- 0.34       | 3.13% +/- 0.50                        |
> | DDS R-50  |   0.64% +/- 0.00    | 0.66% +/- 0.00       | 4.14% +/- 0.05                        |
> | 1-NN CV   |   0.84% +/- 0.05    | 0.84% +/- 0.05       | 3.86% +/- 0.51                        |
> | Logistic  |   0.82% +/- 0.03    | 0.70% +/- 0.15       | 3.36% +/- 0.62                        |
> | PARC      |   0.18% +/- 0.36    | 0.58% +/- 0.17       | 2.93% +/- 0.69                        |
>
> After applying the techniques in our paper, every method tested selects a better model than ResNet-50 w/ ImageNet on average. The improvement is consistent but small because many methods end up choosing ResNet-50 ImageNet roughly half of the time. However, there are cases where the model selected is significantly better than ResNet-50 ImageNet. For instance, when transferring to CUB200, all methods except NCE choose a ResNet-50 model trained on NABirds instead of ImageNet. This provides an over 5% accuracy boost to the final model vs. using the ImageNet one. While this transfer might be obvious to deep learning researchers, being able to make these decisions algorithmically allows these kinds of improvement to be more accessible to those less familiar with the field. Moreover, if we have some computational budget restrictions and, for instance, only allow the methods to select from GoogLeNet models (last column), we observe an even bigger boost when compared to GoogLeNet trained on ImageNet.
>
> We would also like to reiterate that this method of evaluation only considers the top model returned, which does not test a method's ability to calibrate scores across different architectures and tasks (which we know PARC excels at). Robustness to source diversity is a core tenant of this work and is why we use Pearson Correlation for evaluation (among the other reasons described above). The top-1 results you see here are more narrow and don't take all models in the model bank into account.
>
> **PC is hard to interpret.** We definitely hear you on that one. It's hard to intuitively understand small differences in correlation. Hopefully the accuracy when compared to R-50 ImageNet in the table above is easier to interpret. We also plan on adding either this metric or relative accuracy as described in our response to Reviewer YN5P to address this concern in the paper.
>
>
> **Why do we use "Scalable"?** We think that selecting models from a large, diverse source model bank is an important task. The first step to solving this task is to have a method that works on a diverse set of source models while still being fast. Our paper currently focuses on taking this first step, and thus we present a fast method that scales linearly with the size of the model bank, which should be enough to select from 100s of models in a reasonable amount of time. We hope that future papers can tackle the problem of further reducing this to sub-linear scaling (e.g., with some model embedding that allows binary search), but that is outside the scope of this initial look at the problem.
>
> **PARC and the other techniques have no theoretical justification.** This is true, and a theoretical justification would be desirable. However, other theory-backed works such as LEEP seem to underperform with a diverse source model bank. So, in this paper, we focus on creating a method that works empirically first and hope that future work can more closely analyze the theory.
>
>
> **Closing Remarks.** Thank you again for your comments! You're definitely not the only one with these concerns, so we intend to include parts of this response in the final draft of the paper.

---

> ### Public Comment · ~Kaichao_You1 · 2021-12-03
> **significance of this line of research**
>
> Hi, I'm an author from the paper "LogME: Practical Assessment of Pre-trained Models for Transfer Learning". I'm glad you noticed our paper, and would like to explain the significance of transferability estimation.
>
> Indeed we found some larger models tend to perform better, but in practice performance is not the only thing we care. Industrial scenarios would have a limited budget for FLOPS and memory footprint, and therefore we cannot blindly use the largest model. ideally, we should first filter out those models within the computational and memory budget, then ranking these models to select the best, which have about the same complexity.
>
> This viewpoint is explained in our recent work [Ranking and Tuning Pre-trained Models: A New Paradigm of Exploiting Model Hubs](https://arxiv.org/pdf/2110.10545).

---

### Official Review · Reviewer_5SWG · 2021-07-16

**Rating:** 7
**Confidence:** 4

**Summary:**

The paper studies methods to select source models from a diverse set, considering both the quality of the proxy for downstream performance, and the speed of computing the proxy. The paper introduces a new benchmark for evaluating the ability to choose from diverse source models, studies a number of existing methods, and highlights some shortcomings. It then proposes some tweaks to improve these. Finally, the paper proposes a new proxy PARC, which performs well on their benchmark (also using the aforementioned tricks), and also on an extended evaluation on crowd-sourced models.

**Limitations And Societal Impact:**

Limitations are well acknowledged. There is not much discussion of societal impact, however, for this generic ML/CV paper there is no need for this.

**Main Review:**

*Summary*

Overall I think that the paper is well written and organized. It makes a substantial contribution to a relevant practical problem. The experiments look very thorough. Overall, I recommend acceptance, with just a few small comments, that if addressed I might increase my score further.

*Originality*

The task of selecting from different source models has been studied before. However, this work introduces a substantially more systematic study than previous work. The paper also considers practical aspects carefully. I believe this is a significant original contribution. The literature is well cited, one extra relevant paper that studies different proxies that should probably be mentioned is [Which Model to Transfer? Finding the Needle in the Growing Haystack](https://arxiv.org/abs/2010.06402).


*Quality*

Overall, the benchmark and evaluation are thorough and informative. I have no major concerns, a few comments:

- I like the detailed practical considerations: in particular reporting real-world wall clock times. To compliment these timings, it would be useful to also have an analysis of the various methods that is not tied so strongly to the particular hardware used: such as reporting total FLOPs, or even an asymptotic analysis.
- I like the clear ablation of the tricks to improve existing methods, and the core PARC algorithm.
- The additional experiments on detection are a strength, even though PARC is not the best performing algorithm.
- How much does the analysis depend on the correlation measure, would the results be significantly different if Spearman's correlation was reported in Table 1?
- It would be nice to see a summary of which models are actually selected for the different methods, and identify qualitatively their failure modes.
- The paper mentions "Furthermore, practitioners are typically interested in the trade-off between performance and inference speed" -- however, as far as I can tell, the gold standard downstream performance is just accuracy (after finetuning) and does not take into account the speed of the resulting model? (I think this is OK, but needs to be clarified that this aspect is not included).

*Clarity*

The paper is well organized and clear. A couple of minor things
- PARC acronym is used before it is defined
- The notation in Eqns (2) and (5) seems confusing. It seems that correlation is being computed between two unpaired sets, whereas isn't it computed over a set of pairs?
-  typo: "We train all source models and transfers using SGD" -> transfer

*Significance*

I think the PARC method could be useful, although it is not a major extension over existing approaches, I think it is of moderate significance. The benchmark and thorough analysis of existing approaches I think is of high significance. Having a meaningful dataset to analyze and compare methods would be of great value.


POST REBUTTAL

The additional results are appreciated. It is interesting that under Spearman the rank of the methods is quite different. I think this should be acknowledged as a limitation, but I will keep my positive score.


**Time Spent Reviewing:**

5

---

> ### Author Response · Authors · 2021-08-10
> **Author Response to Reviewer 5SWG**
>
> Thanks for your positive comments and feedback! We appreciate your comments on originality, and we'll definitely add [1] to the related work as it seems very relevant.
>
> **Hardware-Independent Evaluation of Speed.** We definitely think real-world performance of these methods is important for this task. While we'd like to compute some notion of FLOPS that would be hardware independent, that is likely not possible with the variety of model selection methods tested here. The operations performed by each method are very different and thus hard to compare (things like matrix multiplication and inversion are possible to profile, but some methods depend on utility functions like np.unique and np.where which are not as easily represented). Since all wall-clock times were benchmarked on the same hardware, we hope that gives enough intuition as to how expensive each method is.
>
> **Would the results change if we used Spearman instead?** We use Pearson Correlation because we want the selection algorithms to be able to produce a score that correlates well with final transfer performance. That is, if a model is x% better than another model, the score output by the model selection algorithm should be some constant times x% higher. We want this property so that our evaluation is applicable to any subset of source models, not just the specific ones in our model bank. While Spearman Correlation is a metric for computing correlation, it only takes into account the relative order of the returned methods and not the actual % difference. This doesn't fully measure what we want it to, but regardless, the results don't seem to change much if we use Spearman instead:
>
> | Method                          |   Pearson           | Spearman        |
> |:--------------------------------|:-------------------:|:---------------:|
> | LEEP (L)                        | 26.83% +/- 0.12     | 29.11% +/- 0.27 |
> | NCE (L)                         | 28.27% +/- 0.47     | 32.32% +/- 0.61 |
> | HScore (L, k=16)                | 65.61% +/- 0.18     | 69.08% +/- 0.31 |
> | RSA R-50 (L, k=32)              | 68.76% +/- 0.04     | **74.01% +/- 0.37** |
> | DDS R-50 (L, k=32)              | 68.25% +/- 0.15     | 71.59% +/- 0.39 |
> | 1-NN CV (L, k=256)              | 68.06% +/- 0.70     | 72.65% +/- 0.78 |
> | Logistic (L, k=256)             | 68.99% +/- 1.27     | 72.52% +/- 1.41 |
> | **PARC** (L, k=32)              | **70.28% +/- 0.47** | 72.21% +/- 0.71 |
>
> RSA breaks out on top using a ResNet-50 model trained on each target, but again, Spearman only considers the absolute ranking of the models and not the calibration of the score like we want. We describe another metric in our response to Reviewer YN5P and plan to add the relative accuracy we define there to the paper to compliment Pearson Correlation.
>
> **What does each method pick?** There's not enough space to list all 48 selections here (we've added it to the appendix), so we'll describe some notable ones. For CUB200 (a bird classification dataset), all methods except NCE choose ResNet-50 pretrained on NABirds with a final transfer accuracy of 85%. NCE, on the other hand, picks ResNet-50 pretrained on ImageNet, which only gets an 80% transfer accuracy. For NABirds, the same methods pick ResNet-50 pretrained on CUB200, which makes sense. However, this time, ResNet-50 pretrained on ImageNet gets around the same accuracy so there's not much difference in results (both are around 77%). For Oxford Pets, the methods are split between picking ResNet-50 trained on Stanford Dogs and ImageNet (with Stanford Dogs being the best choice). This is also true the other way around. These results are pretty interesting and show that these methods can leverage these inherent task similarities, so we'll add this discussion to the paper.
>
> **Other Comments.** As you pointed out, "furthermore, practitioners are typically interested in the trade-off between performance and inference speed" was perhaps an over generalization on our part. We'll reword that to be more specific to practitioners with a computation budget in mind (e.g., those who are working with mobile computer vision or embedded devices). We'll also fix the clarity issues you pointed out. Thanks for bringing all of this to our attention!
>
> [[1](https://arxiv.org/pdf/2010.06402.pdf)] Renggli et al. *Which Model to Transfer? Finding the Needle in the Growing Haystack*. Arxiv 2020.

---

### Official Review · Reviewer_YN5P · 2021-07-20

**Rating:** 5
**Confidence:** 4

**Summary:**

The paper studies model selection for transfer learning: given a number of source or upstream models, and a new downstream or target data, how can we choose the best model to finetune? The paper offers two main contributions. First, it proposes a specific benchmark with source models, target tasks, and an evaluation metric to compare different approaches. Second, after testing on a number of methods from the recent literature, it identifies some algorithmic aspects that seem to hurt these methods and proposes an alternative approach called Pairwise Annotation Representation Comparison (PARC) which addresses those. Experiments suggest PARC outperforms the original algorithms.

**Main Review:**

I'll list strong (+) and weak (-) points, together with questions (?).

(+) I think model selection for transfer learning is indeed of very high practical importance.

(+) The code and benchmark (pre-trained models?) will be open-sourced, thus helping people to use and report the benchmark.

(-) The source datasets (and models) seem fairly small. It would make sense to add models trained on Imagenet (or parts of it).

(-) My main concern is related to the evaluation metric (presented in equation 2). The goal of model selection for transfer is to identify one model to transfer (or, say, 2 or 3 if one has more budget). We only care about the finetuned "quality" of the selected model. However, the correlation approach presented in eq. (2) measures agreement for all the models --including the long tail of "irrelevant" ones. I don't think we need to account (at least so heavily) for those sub-optimal models. If the benchmark has access to the \omega_s^t for all (s, t) --that is, the fine-tuned performance for source model s on target data t--, I think a more reasonable performance metric would be something like: for some small k >= 1, the top-k hindsight performance is the average \omega_s^t over the top-k models s with highest \alpha_s^t scores. In other words, we use the \alpha's to select the models, and then we look at the finetuned performance \omega we would get (we can normalize it somehow, to give all target datasets a similar importance). I'd add two columns (k=1 and k=3 for instance) to the tables in the paper, reporting a metric like this one (other work reports a notion of regret, [1]). In this case, I'm open to revisit my score (no need to remove the current mean PC, but discuss this).

(+) The proposed heuristic where some sort of model complexity (like the second term in equation 6) seems to work well. I wonder if this is indeed equally useful when looking at a metric that focuses on choosing the right model. In any case, it would be nice if this aspect is studied in follow-up work.

(+) PARC is relatively fast to compute, and it doesn't require access to a pre-trained model in the task data (as RSA or DDS do).

(+) I think previous work hasn't been able to properly deal with the fact that different source models may offer representations with very different dimensionality. Using PCA seems like a good idea (and results suggest so too).

(+) PARC can naturally deal with tasks other than classification (as g(.) unlocks the use of any relevant mapping).

Typos:

- line 71: methods.
- line 85: transferability.


[1] - Which Model to Transfer? Finding the Needle in the Growing Haystack (https://arxiv.org/abs/2010.06402)

**Time Spent Reviewing:**

3

---

> ### Author Response · Authors · 2021-08-10
> **Author Response to Reviewer YN5P**
>
> Thank you for your time and valuable feedback! We appreciate your comments on the importance of this task and corresponding open-source benchmark (we will release probe sets, model weights, and evaluation code).
>
> **Adding ImageNet to the source model bank.** We do include ImageNet as a source dataset for all 4 model architectures (see Sec 3.1, Line 135). In the crowd-sourced benchmark, we also include 10 source models trained using different pretext tasks on either ImageNet 1k or ImageNet 22k. Eventually, we aim to have 100s of source models, but chose 65 source models as a first benchmark to highlight the importance of addressing scalable and diverse model selection.
>
> **Is Pearson Correlation a suitable metric?** We believe that Pearson Correlation (instead of some top-k accuracy) is the right metric to use for this task for several reasons:
>  1. We want our results to be meaningful no matter the model bank being used. If we just looked at the top model output, our results would be entirely invalidated if models were added or removed to the model bank. We care about the intrinsic quality of the selection models themselves, rather than specifically how they perform on our exact model bank.
>  2. Our goal is to address *diverse* model selection, meaning that the source models should vary in architecture, dataset, and task. If we only look at the top model returned by the selection algorithm, we're not evaluating how well that method can compare across these source model variations. In order to do that, we need to take the ranking of all the models into account.
>  3. Practitioners using these model selection algorithms will have different needs depending on their situation. If they need to run their models on phones or other low-power devices, ResNet-50 might not be an option. They'd want to compare e.g., ResNet-18, GoogLeNet, and MobileNet among other more optimized architectures, meaning our evaluation method must take that into account.
>
> **Other Metrics.** Nevertheless, as you point out, it would be useful to have additional metrics that depend more on the actual top results predicted by the method. For this purpose, we think relative accuracy would be the easiest to interpret. That is, take the top-k models suggested by the algorithm, fetch their transfer performance and average those numbers like you suggested. Finally, divide this average by the transfer performance of the best model to obtain a notion of "model selection accuracy". This is similar to the metric you propose and that in [1] except that it can be meaningfully averaged over different target datasets. Here are the results for different values of k. We list columns with the original performance of each method (w/o the tricks discussed in the paper), as well as the same with the tricks added (w/ Tricks).
>
> | Method    | Original, k=1   | Original, k=3   | w/ Tricks, k=1  | w/ Tricks, k=3  | w/ Tricks, k=5  |
> |:----------|:---------------:|:---------------:|:---------------:|:---------------:|:---------------:|
> | LEEP      | 93.18% +/- 0.00 | 90.72% +/- 0.04 | 99.56% +/- 0.00 | 92.79% +/- 0.02 | 90.86% +/- 0.00 |
> | NCE       | 95.80% +/- 1.65 | 89.66% +/- 1.62 | 98.82% +/- 0.40 | 96.80% +/- 0.26 | 94.56% +/- 0.25 |
> | HScore    | 84.78% +/- 2.77 | 83.55% +/- 2.17 | 99.46% +/- 0.10 | 98.63% +/- 0.04 | 97.66% +/- 0.14 |
> | RSA R-50  | 98.27% +/- 0.12 | 98.32% +/- 0.15 | 99.35% +/- 0.04 | 98.66% +/- 0.03 | 97.75% +/- 0.11 |
> | DDS R-50  | 99.37% +/- 0.00 | 98.25% +/- 0.17 | 99.37% +/- 0.01 | 98.62% +/- 0.03 | 97.70% +/- 0.09 |
> | 1-NN CV   | 99.60% +/- 0.05 | 97.07% +/- 0.79 | 99.60% +/- 0.05 | 97.75% +/- 0.27 | 96.94% +/- 0.02 |
> | Logistic  | 99.58% +/- 0.03 | 96.46% +/- 0.92 | 99.45% +/- 0.17 | 97.86% +/- 0.26 | 97.08% +/- 0.27 |
> | PARC      |         -       |        -        | 99.31% +/- 0.18 | 98.43% +/- 0.12 | 97.88% +/- 0.00 |
>
> Three points are clear from these results:
>  1. Adding the tricks described in the paper significantly improves the top-k results for almost all methods. This seems to be true no matter what mode of analysis we use (see our response to Reviewer LikY for another metric that shows similar results).
>  2. With the tricks applied, some methods like LEEP have very high top-1 accuracies, but quickly fall off as more than 1 model is taken into account. Top-1 results like these are inherently flawed in that they depend significantly on the list of models used, as small changes in that list can quickly shake up which algorithm performs the best, which is alleviated by a metric like Pearson Correlation which takes all models into account.
>  3. While PARC w/ Tricks isn't explicitly at the top except for k >= 5, it's still a very strong contender that's fast and can be applied to any source task, architecture, or target task. Moreover, it's well calibrated for all types of models (which is what the Pearson Correlation results show).
>
> We'd like to reiterate that this metric only tests 1-5 of the top selected models and does not test whether the algorithm is robust across source architecture or task (as those variations aren't likely to come up in the top 5 models). Thus, we intend to keep Pearson Correlation as our main metric for evaluation, since top-k metrics don't capture our goal of testing a method's robustness to model diversity. Nevertheless, there is merit to this type of analysis, so we'll add these results and corresponding discussion to the final draft of the paper.
>
> **Typos.** Thanks for bringing these to our attention! We've fixed them in the paper.
>
>
> [[1](https://arxiv.org/pdf/2010.06402.pdf)] Renggli et al. *Which Model to Transfer? Finding the Needle in the Growing Haystack*. Arxiv 2020.

---

> > ### Comment · Reviewer_YN5P · 2021-08-24
> > **Scalable Diverse Model Selection for Accessible Transfer Learning**
> >
> > I'd like to thank the authors for their detailed answer. I appreciate it.
> >
> > I'd still favor the top-k metrics as most useful for practical purposes, while I acknowledge the point raised by authors that sometimes not all models are relevant/feasible. However, in that case, again, the end-user's goal is to select the best model among those that apply. Thus, one could report the top-k metric for each of a number of subsets of models that are comparable (same range of FLOPs/runtime, same architecture, etc), as is done in [1].
> >
> > To be clear, I'm not advocating for k=1; probably something a bit more forgiving like k=3 or k=5 makes sense, or if the number of models is huge, it can be k=5% or something like that.
> >
> > In any case, I appreciate the table provided above, and I think it's fair to report all the metrics and let the reader/practitioner decide what's the most appropriate approach for their use-case. There's value in the comparison, and it's interesting to see how different methods perform best with respect to different metrics.
> >
> > I'll raise my score, and won't oppose to acceptance.

---

### Official Review · Reviewer_cgQC · 2021-07-24

**Rating:** 8
**Confidence:** 4

**Summary:**

The paper proposes a model selection method wherein best experts are selected from a model zoo to fine-tune on the target task. The proposed method "Pairwise Annotation Representation Comparison" is an improvement over RSA [8], instead of using a small dnn trained on target task (or probe network) the authors use spearman correlation between RDM of features and label as model selection score (called PARC score). The proposed method is compared to many baselines NCE, LEEP, RSA, DDS etc. and shows better correlation to fine-tuning accuracy.

**Limitations And Societal Impact:**

Limitations are addressed in detail in Sec. 7.


**Main Review:**

Strengths:
+ The proposed method is compared to many different baselines. In Tab. 2, 3 and 4 the proposed method shows better correlation to fine-tuning accuracy compared to RSA, LEEP, NCE, Nearest Neighbors model selection baselines. Tab. 2 also performs an ablation of the correlation score when architecture, source dataset and target dataset are changed.
+ The approach is extended to show that selection with the same scoring function works for object detection tasks. Model selection for object detection is not studied in any of the prior arts.

Weakness:
- The results on transfer between different models to the target task are missing. E.g. does imagenet trained expert or some of the candidate architecture always work better than the rest?
- Some other relevant baselines to the proposed method which are not discussed are below:
1. Cui et al "Large Scale Fine-Grained Categorization and Domain-Specific Transfer Learning" -- Uses EMD distance between average class vectors of source, target task to estimate transferability
2. Achille et al "Task2Vec: Task Embedding for Meta-Learning" -- Estimates an embedding space for tasks. If source and target task are close in the embedding space, the transfer is better
3. Deshpande et al "A linearized framework and a new benchmark for model selection for fine-tuning" -- Proposes a Label Feature Correlation method for model selection that uses correlation between label/feature similarity matrix. The score function is similar to proposed approach of Annotation (i.e. label), Representation (i.e. feature) Comparison.


**Time Spent Reviewing:**

5

---

> ### Author Response · Authors · 2021-08-10
> **Author Response to Reviewer cgQC**
>
> Thank you for your review and positive comments! We've incorporated your suggestions:
>
> **Missing Oracle Performance.** We appreciate your request for ground truth results for each transfer and have added a table with the 423 oracle transfer performance results to the Appendix.
>
> **Other Baselines.** Thanks for bringing these works to our attention. We've added a thorough discussion of all of these works to the paper:
>  1. Cui et al. [1] propose a graph-based solution to finding the distance between tasks. In order to do this, they set up a graph with source class feature vectors and target class feature vectors as nodes. This doesn't quite work for our task because they assume they have access to the source class features, which in our case might not even exist (e.g., if the model was trained in a self- or weakly-supervised way). Furthermore, the solution is quite slow and scales poorly with the number of classes in the source and targets. Thus, this work doesn't meet the criteria of source diversity or scalability that we require. Definitely a relevant paper to cite, though.
>  2. Achille et al. [2] propose a task embedding space to find similarities between tasks. They get these task embeddings by computing the Fisher Information Matrix of a probe network. While powerful, they make two important assumptions: that all source models are using the *same architecture* (in fact the same exact model with only the classifier re-trained) and that you have a probe model trained on the target dataset. The latter assumption is not that big of an issue because they only retrain the classifier, and so it has a similar time complexity as the Logistic baseline in our paper. However, both assumptions violate the diversity requirement as the source models are constrained to one architecture and the target task is constrained to be classification. Nevertheless, if this method could be made more diverse, it definitely has a shot at being one of the most scalable, as the task embeddings only need to be computed once per source model and never again.
>  3. Deshpande et al. [3] propose perhaps what is most directly applicable to our work. Their setting is very similar, though not as diverse and with no restrictions on evaluation speed. As far as we can tell, Label-Feature Correlation (LFC) in [3] derives the concept of correlating labels and features using prior work's bounds on generalization performance. We, on the other hand, derive PARC by taking an existing method that works well (RSA) and extending it to use labels. Both methods operate on a similar principle, but the resulting equations are different. LFC as described in [3] does seem to be one of the best methods tested, corroborating the strength of comparing labels to features. Because [3] doesn't lean too heavily on having a variety of architectures (they only test 2 as far as we can tell) or arbitrary crowd sourced models (they train all of their models themselves), we view it to be largely complementary to this work.
>
> We'll definitely these works to the discussion and related work of our paper. Thanks again for recommending them!
>
>
> [[1](https://arxiv.org/pdf/1806.06193.pdf)] Cui et al. *Large Scale Fine-Grained Categorization and Domain-Specific Transfer Learning*. CVPR 2018.
>
> [[2](https://arxiv.org/pdf/1902.03545.pdf)] Achille et al. *Task2Vec: Task Embedding for Meta-Learning*. ICCV 2019.
>
> [[3](https://arxiv.org/pdf/2102.00084.pdf)] Deshpande et al. *A linearized framework and a new benchmark for model selection for fine-tuning*. Arxiv 2021.

---

### Official Review · Reviewer_rwfu · 2021-08-01

**Rating:** 5
**Confidence:** 4

**Summary:**

The authors introduce the “scalable diverse model selection” task and introduce several tools and benchmarks for evaluating model selection methods in this setting. They show that current transferability and model selection methods fail to beat simple baselines in this new setting. They analyze the reason for this case and provide techniques to improve performance. They develop PARC, a method that outperforms other methods on diverse model selection.

**Limitations And Societal Impact:**

Yes

**Main Review:**

Pros:
- The paper is well written and it is a timely paper to discuss the limitations of existing model selection methods, such as they don’t consider different model architectures. The comparison of different methods regarding probe size, architectures, and feature dimension might be interesting to readers in this area.


Cons:

- The authors compare several model selection algorithms, i.e., probabilistic methods and feature-based methods. However, those algorithms are not introduced anywhere and the authors may assume that readers are familiar with those algorithms.
- My major concern is that the proposed benchmark can not fully evaluate “scalable” model selection and it is not clear how to scale the benchmark. The authors collected 8 source datasets to 6 target datasets across 4 different commonly used architectures. However, the size of the source datasets and target datasets in total is 32, which is not quite big. The architectures used (ResNet-50/18, GoogleNet, AlexNet) are not quite representative. Architectures with different input resolutions would be great to add. The target 6 datasets are also quite small, among which their performance is almost saturated, such as the accuracy of Pets/CIFAR10 is above 95%. There are Google VTAB datasets (19) which could be much larger as the target sets.

- It is not clear why the new proposed PARC has an advantage over existing methods such as RSA. The PARC seems pretty similar with the Label-Feature Correlation proposed in [1], which would be great to include for discussion.

- It would be clearer if the authors can make clear what exactly are the models and datasets described in section 6.1. It is referenced but not stated anywhere.

- It is not clear how the benchmarks can be released with protocols and facilitate future comparisons.

[1] A linearized framework and a new benchmark for model selection for fine-tuning, arXiv 2021

**Time Spent Reviewing:**

10

---

> ### Author Response · Authors · 2021-08-10
> **Author Response to Reviewer rfwu**
>
> Thank you very much for your comments! You've brought up some very reasonable concerns that we'll attempt to address here.
>
> **Prior work isn't explained.** We tried to introduce the relevant aspects of each method in Sec. 3.2, though sadly we did not have enough space to do a more in-depth explanation. We will try to make room for a more detailed dive into the prior methods for the final submission.
>
> **The benchmark isn't big enough to test scalability.** Thank you for your suggestion to use the Google VTAB datasets. Sadly, we do not have time during the rebuttal to train additional models to increase the size of the benchmark. However, we hope the crowd sourced benchmark presented in Tab. 5 and Sec. 6.1 can offer a suitable middle ground. It includes 65 source models and 423 total transfers. Importantly, these source models come from arbitrary domains. They contain models trained with different datasets, tasks, and even image sizes (like you suggested). We agree that the ultimate goal will be to select from a set of 1000s of source models and plenty of target datasets. This first look into scalable diverse model selection highlights issues with current benchmarks and evaluation settings and subsequently limitations of current approaches. We hope future and community collaboration can grow this benchmark over time.
>
> **PARC vs. RSA.** From the benchmark results, PARC is slightly better than RSA while not requiring training an additional model like RSA does. The end goal for scalable diverse model selection is to be able to upload some sample of data and receive a pretrained model in a reasonable amount of time. If we have to train an additional model on that data as a proxy, that both slows things down and biases the selection algorithm (see poor results for RSA-Alexnet in Tab. 1). PARC completely avoids that problem while performing as well or better than RSA.
>
> **PARC vs. Label-Feature Correlation.** Thanks for pointing out the similarity between these two works! As far as we can tell, Label-Feature Correlation (LFC) in [1] derives the concept of correlating labels and features using prior work's bounds on generalization performance. We, on the other hand, derive PARC by taking an existing method that works well (RSA) and extending it to use labels directly. Both methods operate on a similar principle, but the resulting equations are different. LFC as described in [1] does seem to be one of the best methods tested, corroborating the strength of comparing labels to features. Because [1] doesn't lean too heavily on having a variety of architectures (they only test 2 as far as we can tell) or arbitrary crowd sourced models (they train all of their models themselves), we view it to be largely complementary to this work. We'll definitely add this to the discussion and related work of our paper. Thanks again for recommending it!
>
> **Models and Datasets described in Sec. 6.1.** We couldn't fit the wide variety of models and architectures in the text of the paper, so we put it in Sec. 4 and Tab. 6 in the appendix (page 8 of the supplement). If we have space, we'll try to add a summary in to the text of the paper or otherwise make the pointer to the appendix more clear.
>
> **Releasing the benchmark.** The primary method we'll use to release the benchmark is to release the pre-extracted probe sets (along with the evaluation code). This way, we don't actually need to release any data or model weights, as we'll just be releasing features and probabilities. Nevertheless, if we do need to release the models, we'll release the ones we trained ourselves and provide links to the original source for any downloaded models.
>
>
> [[1](https://arxiv.org/pdf/2102.00084.pdf)] Deshpande et al. *A linearized framework and a new benchmark for model selection for fine-tuning*. Arxiv 2021.

---

> > ### Comment · Reviewer_rwfu · 2021-09-03
> > **My concern still remains**
> >
> > Thanks for the detailed reply. I don't quite agree with the claim of the contribution as "We introduce Scalable Diverse Model Selection, a new task that intends to make deep learning for computer vision more accessible". The model selection task itself is not new and a large scale model selection experiment cannot claim the task or method as "new" or "salable". Just like we cannot claim a classification task with 1000 classes or a NN search task with 1000 data as a new task. It is probably more accurate to describe it as a large-scale experiment or benchmark involving more choices of architectures/models than existing method. But similar scaled experiments was also done before such as in [1], which tested on 8 source domains, 8 target datasets with 2 architectures and 3 learning rates.
> >
> > My concern is that the "scalablility" is not well defined and it is not clear whether it describes the benchmark or method. I would expect the authors to show existing method fails to "scale" for large-scale model selection due to either long evaluation time or low accuracy while their proposed method is more efficient or accurate. However, from Table 4 and 5, I can only conclude the proposed method is marginally more accurate but not more "scalable". Due to the confusions, I am prone to keep my score.

---

> > > ### Author Response · Authors · 2021-09-03
> > > **Clarifying our claims of scalability**
> > >
> > > We appreciate the reviewer’s concern and would like to further clarify. We believe the problem setting of model selection among a large number of models which may be trained in diverse ways across distinct architectures is important — we called this scalable and diverse model selection to distinguish our focus from many prior works which studied selection from homogeneous architectures across different source datasets (RSA, DDS), or require a substantial amount of time to process as the number of models increase (H-score). Though some prior methods can technically operate under these conditions, we found them to perform very poorly in practice (LEEP, NCE) and substantially lower than simple baselines (NN, Logistic). We presented this framing as a contribution of our work since we provide a more direct assessment of methods and compose a collection of models which test the properties of scalability and diversity of model selection algorithms. We understand and acknowledge that our model bank could be further increased to include more models, but do believe that issues due to poor scalability can still be highlighted using our benchmark as is.
> > >
> > > For instance, RSA and DDS require ~3 hours of extra training time per target dataset (and prior knowledge of the best architecture), H-Score scales with the cube of the feature size, and the Logistic baseline takes ~10x the time to evaluate as other methods. For these reasons (which we list in Table 1), we claim that PARC is more applicable to scalable model selection than these methods. However, we do not intend to claim that PARC is more scalable than LEEP, NCE, or the other baselines because, in fact, PARC scales at the same rate as those methods (i.e., linearly) and has similar evaluation time. However, the goal of this paper is to address scalability and diversity *simultaneously*, and LEEP and NCE as shown in Table 1 are not robust to diverse source models. Thus, the only methods in our study that are capable of addressing both scalability and diversity at the same time are the Nearest Neighbors baseline (which has been omitted from most prior studies including RSA, DDS, LEEP, NCE, H-Score, and [1]) and PARC, with PARC performing better in every benchmark (Tables 4, 5, 6).
> > >
> > > Finally, we thank the reviewer for bringing up this concern and would be happy to revise our paper to clarify our claim that we "systematically study" (rather than introduce) this setting and find that existing approaches are not viable either due to poor performance or long preparation or evaluation time.

---

### Public Comment · ~Kaichao_You1 · 2021-12-03
**Refer to a related work**

Hi, I'm an author from the paper "LogME: Practical Assessment of Pre-trained Models for Transfer Learning". I'm surprised to find this paper does almost the same thing as the LogME paper but the latter is not mentioned in related works.

Meanwhile, I noticed that there is a discussion about the use of pearson coefficient. I kindly suggest the authors have a look at the weighted Kendall's tau coefficient used in our paper, which is based on rank and more robust.

In my humble opinion, I don't quite understand the motivation of the proposed PARC method. The LogME paper has a clear motivation and its theoretical support is later provided in "Ranking and Tuning Pre-trained Models: A New Paradigm of Exploiting Model Hubs". It would be better if the motivation behind PARC can be better illustrated. I look forward to hearing this poster presentation at NeurIPS :)

---

> ### Public Comment · Authors · 2021-12-04
> **Thanks for pointing that out**
>
> Hello, and thanks for the comment.
>
> It seems that we have missed the citations for both LogMe and OTCE when incorporating reviewer feedback for the camera ready. Thank you for bringing this to our attention. We'll add both citations in the v2 of our Arxiv release.
>
> As for the motivation behind PARC, we consider it mostly as a simple architecture-independent extension to RSA. Our primary motivation for the paper was the benchmark and task, not the method. Hence, why we named the paper after the task and have released our benchmark in an easy to evaluate on format. While developing this work, we noticed that none of the state-of-the-art methods outperformed simple baselines like nearest neighbors, so we spend a good bit of the paper trying to figure out why that's the case. Then we propose a very simple extension to RSA using these findings that we found to be powerful empirically.
>
> While we were working on this project, several good methods such as yours and others have come out. If we were starting today, we would have benchmarked LogMe as well and likely would not have had to introduce PARC.
>
> Hopefully that can clear some of our motivations up. Also thanks for the suggestion to check out weighted Kendall's tau coefficient. We look forward to further discussing this with you at our poster session.

---

### Decision · Program_Chairs · 2021-09-27

**Decision:**

Accept (Poster)

**Comment:**

In summary, the reviewers rate this paper as follows:
* 3 of 5 reviewers view it as being close to the acceptance threshold (2 of 5 slightly below, 1 of 5 slightly above, not updated from "5" to "6" in the official rating, but in a discussion comment).
* 2 of 5 reviewers recommend the paper for acceptance (with scores "7" and "8").
So the reviewers overall are leaning towards acceptance with no reviewer opposing acceptance.

The paper discusses several methods for model selection and ranking and proposes a new method (PARC) for this task. Experiments compare the methods on a new benchmark created for that purpose.

Highlights of the paper mentioned by the reviewers include:
* The presented study is enlightening and thorough.
* The subject is of high practical importance.
* Open-sourcing the benchmark will be valuable.
* The paper is well written.
* The authors' responses addressed several of the reviewers' initial concerns.

Some concerns expressed by the reviewers include the following:
* Limited improvements of the introduced method; similar performance of other approaches.
* Some over-claiming regarding novelty of the approach.
* A lack of the definition of "scalability".
* The value of ranking models vs selecting best models is debated.
* Some related work is missing or could be discussed more appropriately.

In case the paper is published, it would be appreciated if the authors could address the concerns as far as possible. Specifically, the results presented in the authors' responses would be valuable to include.

Overall, I would recommend the paper for acceptance as a poster.